# mTORC1 controls Golgi architecture and vesicle secretion by phosphorylation of SCYL1

Stéphanie Kaeser-Pebernard [1], Christine Vionnet[1], Muriel Mari [2,3], Devanarayanan Siva Sankar[1], Zehan Hu [1], Carole Roubaty[1], Esther Martínez-Martínez [1], Huiyuan Zhao[4], Miguel Spuch-Calvar [5,6], Alke Petri-Fink [5,7], Gregor Rainer [4], Florian Steinberg[8], Fulvio Reggiori [2,3,9] & Jörn Dengjel [1] ✉

The protein kinase mechanistic target of rapamycin complex 1 (mTORC1) is a master regulator of cell growth and proliferation, supporting anabolic reactions and inhibiting catabolic pathways like autophagy. Its hyperactivation is a frequent event in cancer promoting tumor cell proliferation. Several intracellular membrane-associated mTORC1 pools have been identified, linking its function to distinct subcellular localizations. Here, we characterize the N-terminal kinase-like protein SCYL1 as a Golgi-localized target through which mTORC1 controls organelle distribution and extracellular vesicle secretion in breast cancer cells. Under growth conditions, SCYL1 is phosphorylated by mTORC1 on Ser754, supporting Golgi localization. Upon mTORC1 inhibition, Ser754 dephosphorylation leads to SCYL1 displacement to endosomes. Peripheral, dephosphorylated SCYL1 causes Golgi enlargement, redistribution of early and late endosomes and increased extracellular vesicle release. Thus, the mTORC1-controlled phosphorylation status of SCYL1 is an important determinant regulating subcellular distribution and function of endolysosomal compartments. It may also explain the pathophysiology underlying human genetic diseases such as CALFAN syndrome, which is caused by loss-of-function of *SCYL1*.

The mammalian/mechanistic target of rapamycin complex 1 (mTORC1) is a major serine-threonine kinase complex acting at the crossroads between sensing nutrient availability and controlling cellular homeostasis. mTORC1 is well-characterized for its pivotal role in integrating information on amino acid and growth factor levels, positively regulating cell growth and anabolic processes in times of plenty,

while inhibiting catabolic pathways, including autophagy-mediated lysosomal degradation[1,2]. Starvation or drug-induced mTORC1 inactivation lead to inhibition of cell growth and activation of autophagy[3]. Conversely, hyperactivation of mTORC1 signals excessive cell growth and proliferation. It is a frequent event in cancer, with an estimated average occurrence of 70% in all cancer types[4]. mTORC1 aberrant

[1]Department of Biology, University of Fribourg, 1700 Fribourg, Switzerland. [2]Department of Biomedical Sciences of Cells and Systems, University of Groningen, University Medical Center Groningen, 9713 AV Groningen, The Netherlands. [3]Department of Biomedicine, Aarhus University, 8000 Aarhus, Denmark. [4]Section of Medicine, University of Fribourg, 1700 Fribourg, Switzerland. [5]Adolf Merkle Institute, University of Fribourg, 1700 Fribourg, Switzerland. [6]CINBIO, Universidade de Vigo, 36310 Vigo, Spain. [7]Department of Chemistry, University of Fribourg, 1700 Fribourg, Switzerland. [8]Center for Biological Systems Analysis, University of Freiburg, 79104 Freiburg, Germany. [9]Aarhus Institute of Advanced Studies (AIAS), Aarhus University, 8000 Aarhus, Denmark. ✉e-mail: joern.dengjel@unifr.ch

activation is a tumor driver[5], promoting formation, proliferation, and/or invasion, depending on activation timing and cancer type (reviewed in ref. [6]). Although several pharmacological mTORC1 inhibitors have been developed and tested in anti-cancer studies, resistance phenomena are frequently observed, which are in turn addressed by combinatorial treatments[7]. Developing efficient therapeutic strategies against mTORC1 overactivation in cancer will therefore require a deeper understanding of this kinase complex and its downstream effectors and functions.

mTORC1 regulates its downstream targets at specific subcellular locations ensuring precise control of cell growth. Its predominant recruitment site are lysosomal membranes, where it can be inactivated amongst others upon amino acid starvation[8–11]. In yeast *Saccharomyces cerevisiae*, late endosomal TORC1 was identified as a spatially distinct pool responsible for autophagy regulation, as opposed to the vacuolar/lysosomal pool responsible for modulation of translation[12–14]. In mammalian cells, mTORC1 has also been detected at late endosomes[15], in mitochondrial fractions[16–18], and at the Golgi, to which it is recruited by RAB1A and the SLC transporter SLC36A4/PAT4[19,20]. Golgi mTORC1 appears to be regulated independently of the lysosomal pool[20], one function being the regulation of unconventional protein secretion via GRASP55[21]. Regulation of metabolism by mTORC1 therefore seems to be intimately linked to multiple membranous compartments, including those of the secretory pathways, the canonical one leading to the secretion of signal peptide containing proteins, which reach the plasma membrane via ER insertion and COPII-coated vesicle-mediated transport via the Golgi[22]. How mTORC1 regulates distinct (un)conventional secretion pathways at the molecular level, including the secretion of extracellular vesicles (EVs), is unknown.

In an effort to identify mTORC1 effector proteins, responding to its inactivation by dynamic changes in their phosphorylation status, we previously performed a quantitative phosphoproteomic screen in epithelial MCF-7 breast cancer cells and identified SCYL1 as a strong candidate[23]. The N-terminal kinase-like protein SCYL1 maintains Golgi structure and functions by promoting COPI-coated vesicle-mediated retrograde transport from Golgi to ER[24–27]. *Scyl1^mdf* mutant mice display motor neuron disorders recapitulating most human amyotrophic lateral sclerosis (ALS) symptoms[28]. Moreover, loss of function mutations of human *SCYL1* are associated with multiple recessive hereditary disorders that have been grouped under the term cholestasis, acute liver failure, and neurodegeneration (CALFAN) syndrome[29–33]. The syndrome has also been recently associated with recurrent respiratory failure[33]. In addition, SCYL1 has been linked to breast cancer progression, but its precise oncogenic function is disputed[34,35]. Given its subcellular localization and the affected tissues, these observations suggest an important role for SCYL1 in the function of active secretory cells. SCYL1 has no measurable kinase activity[24], oligomerizes via its central HEAT repeats, and interacts with class II ARF GTPase receptors and multiple subunits of the COPI vesicle coat[26]. Though not a golgin itself, SCYL1 colocalizes with protein markers of the ER-Golgi intermediate compartment (ERGIC), cis-Golgi and trans-Golgi network (TGN)[25]. To the latter, it is recruited by the RAB6-binding protein GORAB via its N-terminal moiety[36]. SCYL1 is therefore thought to function as scaffold protein linking different Golgi and ERGIC compartments and COPI-coated vesicles. Strikingly, SCYL1 was previously identified as an upstream AKT-mTORC1 activator in an RNAi screen for kinases involved in basal autophagy[37], demonstrating an important link between SCYL1 and the mTORC1 signaling pathway, and raising the interesting possibility that SCYL1 and mTORC1 are part of a regulatory feedback mechanism.

Here, we uncover an mTORC1-dependent regulatory mechanism of secretory and endolysosomal trafficking pathways relying on SCYL1 phosphorylation. Under growth conditions, mTORC1 directly phosphorylates SCYL1 at Ser754, supporting its localization at the Golgi and proper vesicular transport. Reversely, absence of SCYL1, mTORC1 inhibition, or expression of a SCYL1 phospho-null mutant, all lead to a strong dysfunction of the secretory and endolysosomal pathways, manifested by Golgi enlargement, relocalization of endosomal RAB GTPases, altered lysosome distribution and increased EV secretion. All of these phenotypes can be rescued by ectopic expression of a phospho-mimicking SCYL1 mutant. We therefore demonstrate that, via SCYL1 phosphorylation, mTORC1 controls intracellular transport routes and EV secretion.

## Results

### mTORC1 inhibition triggers a redistribution of SCYL1

Our initial study identified SCYL1 as a putative mTORC1-responsive protein in epithelial MCF-7 breast cancer cells[23]. To test whether changes in mTORC1 activity commonly regulate the distribution of SCYL1, MCF-7 cells were compared to two additional epithelial cancer-derived cells (HeLa and A549), as well as to one immortalized (non-cancerous) epithelial cell line (hTERT-RPE-1). All lines were treated for 3 h with rapamycin, or starved in Hank's balanced saline solution (HBSS), and endogenous SCYL1 localization was monitored by immunofluorescence (Fig. 1a, b and Supplementary Fig. 1a). Under growing conditions, endogenous SCYL1 remained predominantly in a perinuclear, semi-compact area, with a fraction of SCYL1 present in discrete cytosolic puncta, distal from the nucleus and Golgi (Fig. 1a, b and Supplementary Fig. 1a–d). Upon pharmacological and physiological mTORC1 inhibition, we observed an evident increase in the proportion of peripheral cytosolic SCYL1-positive puncta (Fig. 1a, b and Supplementary Fig. 1a–d), suggesting that SCYL1 localization is indeed influenced by mTORC1 activity. These regulated changes in subcellular SCYL1 distribution were visible in all cell lines tested (Fig. 1a, b and Supplementary Fig. 1a–d). SCYL1 foci distance to the nucleus increased in all cell lines, most significantly in MCF-7 and HeLa cells (Fig. 1b). Hence, SCYL1 redistribution in response to mTORC1 inhibition seems to represent a general phenomenon in cells of epithelial origin.

To proceed in studying the role of SCYL1 in the most relevant cell model, we analyzed mRNA and protein levels in commonly used cell lines using public databases (https://www.proteinatlas.org/; https://www.proteomicsdb.org/). The highest levels of both mRNA and protein were found in breast and mammary gland-derived cell lines, among them MCF-7 cells, which is in line with the observations that loss-of-function mutations of *SCYL1* lead to phenotypes in secretory organs[29–33]. MCF-7 cells are considered poorly aggressive with a low metastatic potential. They conserved their secretory capacity, since their culture in egg white stimulates the development of acini and duct-like structures that support the secretion of beta-casein (reviewed in ref. [38]). Hence, we chose this cell line to study the potential crosstalk between mTORC1 and SCYL1-dependent secretory functions[23,37].

SCYL1 being reported as predominantly located at the ER-Golgi interface, we monitored its distance to Golgi in normal and mTORC1-inhibited MCF-7 cells. Whereas SCYL1 closely associated with cis-Golgi marker GM130/GOLGA2 in untreated cells, a fraction of SCYL1 proteins rapidly delocalized to cytosolic locations upon rapamycin treatment, and SCYL1 foci distance to the Golgi surface significantly increased (Fig. 1c–f and Supplementary Fig. 1a–d).

To substantiate these data biochemically, we performed a cell fractionation by differential centrifugation and analyzed the 17 K pellet, which is enriched in membranes of the Golgi and endolysosomal compartments[39], in growing and mTORC1-inhibited conditions. Rapamycin and starvation treatments increased the amount of SCYL1 in the 17 K pellet, whereas the total amount of endosomes did not vary, as visualized by stable levels of early endosome protein EEA1 and the late endosome marker RAB7 (Fig. 1g–h). Also, the total SCYL1 protein pool remained unaltered (Fig. 1g–h, WCL data), suggesting that mTORC1-specifically regulates SCYL1 association to membrane compartment(s) enriched in endosomes/lysosomes. Treatment with concanamycin A

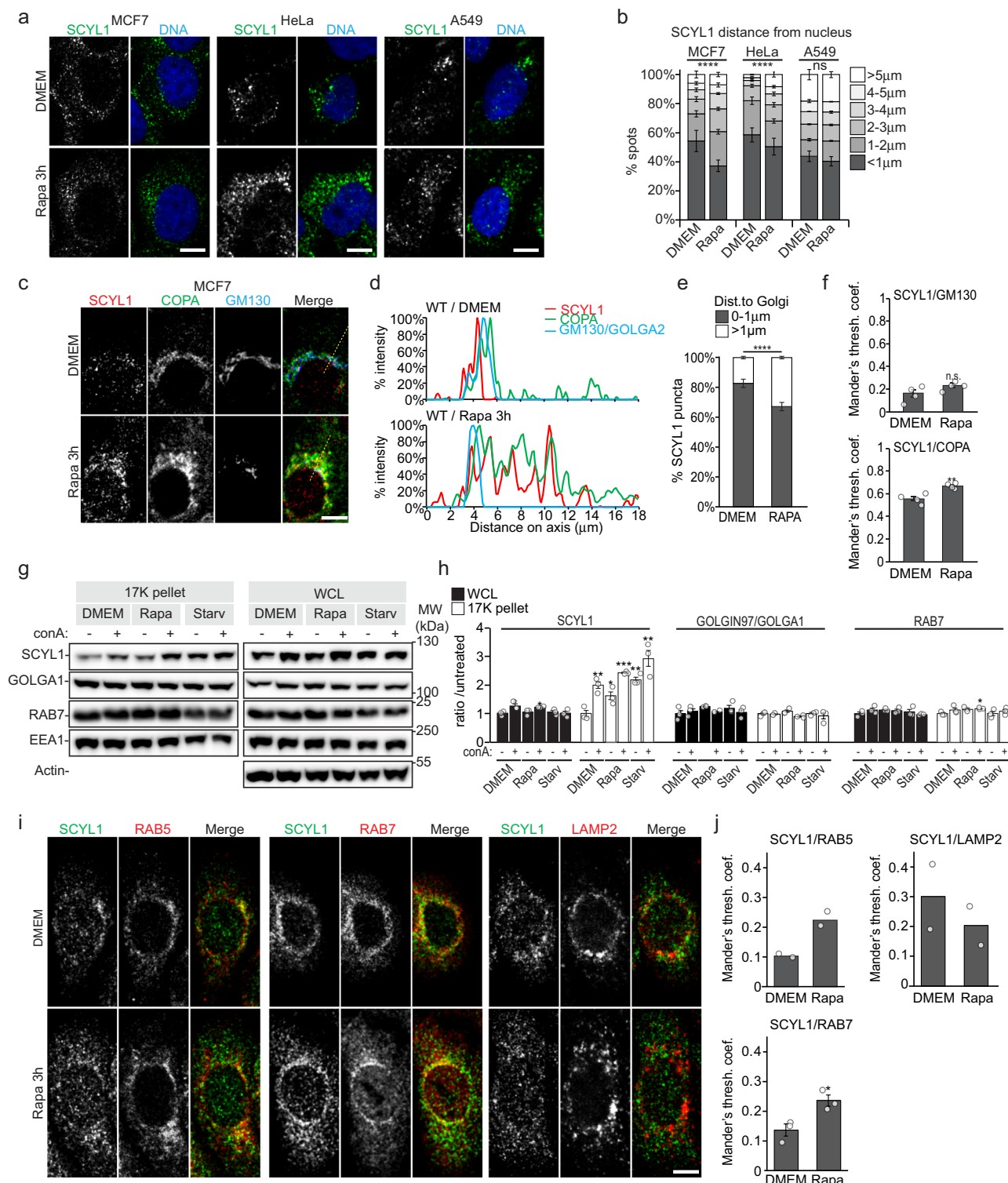

(ConA), an inhibitor of V-ATPase blocking lysosomal degradation, stabilized SCYL1 in the 17 K fraction, indicating that a part of membrane-associated SCYL1 underwent lysosomal degradation (Fig. 1g–h). We conclude that physiological and pharmacological mTORC1 inhibition leads to SCYL1 relocalization to a different sub-cellular membraneous compartment.

### mTORC inhibition leads to SCYL1 endosomal localization

To determine the nature of cytosolic SCYL1-positive compartments detected upon mTORC1 inhibition, we colocalized SCYL1 with specific organelle marker proteins in MCF-7 cells by immuno-fluorescence (Fig. 1c–f, i–j). Increased cytosolic staining did not lead to an obvious reduction of Golgi-localized SCYL1 as indicated by co-localization with GM130 (Fig. 1c, d, f). Interestingly, the COPI-coat subunit COPA, a known SCYL1 interactor[24], localized with SCYL1 to the cell periphery upon rapamycin treatment, and SCYL1/COPA co-localization increased upon treatment, indicating that mTORC1 inhibition supports localization of SCYL1 to COPI-coated vesicles (Fig. 1c, d, f). More strikingly, SCYL1-positive early and late endo-somes as indicated by co-localization with the marker proteins

**Fig. 1 | mTORC1 inhibition leads to SCYL1 redistribution to peripheral cytoplasmic locations. a** SCYL1 localizes to cell periphery upon mTORC1 inhibition. Endogenous SCYL1 puncta localize further away from nuclei upon rapamycin treatment. Representative fluorescence microscopy images from MCF-7, HeLa and A549 cells in untreated (DMEM) and Rapamycin-treated (Rapa, 3 h, 100 nM) conditions. One representative image out of $n > 3$ biological replicates is shown. Scale bar = 10 μm. See also Supplementary Fig. 1. **b** Quantification of SCYL1 puncta distribution shown in **a**. Distances of individual SCYL1 puncta from the closest nuclear envelope were averaged per image and their relative distribution is shown. An unpaired two-tailed Student's *t*-test was used to compare DMEM and Rapa-treated cells: ****$p \leq 0.0001$ (MCF-7: $p = 1.25e$-19; HeLa: $p = 2.87e$-68; A549: $p = 0.84$). $n = 3$ biological replicates, with a minimum of 10 cells measured per replicate. Error bars, SEM, ns: not significant. **c** The COPI-coat complex redistributes with SCYL1 to cytoplasmic locations in mTORC1-inhibited conditions. Fluorescence micrograph from WT MCF-7 cells untreated (DMEM) or treated for 3 h with 100 nM Rapa. Yellow lines: line drawings used to generate plot profiles in **d**. Representative pictures of $n = 3$ biological replicates are shown. Scale bar = 10 μm. **d** Plot profiles of **c**. **e** SCYL1 distance to Golgi increases upon Rapamycin treatment. SCYL1 puncta distance to GM130-positive Golgi edge as in **c** was measured and compared between DMEM and 3 h Rapamycin-treated conditions; unpaired two-tailed Student's *t*-test, ****: $p = 3.91E$-181. $n = 8$ biological replicates for DMEM and $n = 7$ for Rapa (SCYL1 foci from >10 cells measured per replicate). Error bars: SEM. **f** Co-localization quantification of **c**. An unpaired two-tailed Student's *t*-test was used to compare co-localization changes (SCYL1/GM130: $p = 0.14$ (ns); SCYL1/COPA: $p = 0.0038$ (**)). $n = 4$ biological

replicates. Error bars: SEM. **g** Endosomal SCYL1 increases upon mTORC1 inhibition. Cellular fractionation of MCF-7 cells untreated (DMEM) or treated for 3 h with 100 nM Rapa or starved in HBSS (Starv). ConA was added in parallel to all conditions to block lysosomal degradation. Proteins within each fraction were extracted and quantified, and equal amounts were loaded per well. WCL: Whole-cell lysates; 17k: 17,000 x *g* fractionation pellet enriched in endosomal membranes. One representative experiment out of $n = 3$ biological replicates is shown. **h** Quantification of **g**. $n = 3$. *p*-values: unpaired two-tailed Student's *t*-test value of the indicated data point, compared with the reference–value "DMEM-conA". WCL SCYL1 protein levels were normalized to ß-Actin/ACTB levels, whereas 17 K SCYL1 protein levels were normalized to EEA1 levels. *p*-values: SCYL1/WCL, DMEM + conA $p = 0.044$ (*); Rapa + conA $p = 0.011$ (*). SCYL1/17 K: DMEM + conA: $p = 0.0073$ (**); Rapa – conA: $p = 0.045$ (*); Rapa + conA: $p = 0.00044$ (***); Starv – conA: $p = 0.0021$ (**); Starv + conA: $p = 0.0058$ (**). RAB7/17 K: Rapa + conA: $p = 0.012$ (*). Error bars: SEM. **i** SCYL1 co-localization with early and late endosome markers RAB5 and RAB7 increases upon mTORC1 inhibition. Fluorescence micrograph from untreated WT MCF-7 cells (DMEM) or treated for 3 h with 100 nM Rapamycin ("Rapa"). Representative pictures of $n = 3$ replicates are shown. Scale bar = 10 μm. **j** Co-localization quantification of **i**. $n = 2$ biological replicates for SCYL1/RAB5 and SCYL1/LAMP2 measurements, and $n = 3$ for SCYL1/RAB7 measurements. An unpaired two-tailed Student's *t*-test was used to compare SCYL1/RAB7 co-localization changes between DMEM and Rapa-treated conditions ($p = 0.025$ (*)). Ten cells were measured per slide, and three slides averaged per biological replicate. Error bars: SEM. All source data are provided as Source Data File.

RAB5 and RAB7, respectively, shifted from the peri-nuclear area in growth conditions to the cell periphery under rapamycin treatment (Fig. 1i–j). As this was not the case for LAMP2-positive lysosomes, it might indicate a change in endosome positioning, which does not affect the entire vesicular compartment (Fig. 1i–j). To substantiate this observation, SCYL1 co-localization with the early endosomal marker protein sorting nexin 1 (SNX1) was evaluated in MCF-7 and RPE-1 cells. In untreated cells, SCYL1 puncta around the Golgi were frequently SNX1-positive (Supplementary Fig. 1c–g). In contrast, rapamycin treatment increased the co-localization between SCYL1 and SNX1 in the cell periphery in both cell lines (Supplementary Fig. 1c–g). Altogether, these observations suggest that mTORC1 inactivation triggers the localization of SCYL1 to peripheral early and late endosomes.

## SCYL1 KO perturbs Golgi and endolysosomal compartments

To understand the role of SCYL1 in mTORC1-mediated regulation of cell homeostasis, we took advantage of CRISPR/Cas9 to generate *SCYL1* knock-out MCF-7 cell lines (*SCYL1* KO). Two *SCYL1* KO clones targeting two distinct genomic sequences were obtained, and western blotting confirmed the ablation of SCYL1 (Fig. 2a). As *SCYL1* RNAi cells display an enlarged Golgi and reduced mTORC1 activity[24–26,37], we examined whether *SCYL1* KO recapitulates these phenotypes. *SCYL1* KO cells consistently presented an enlarged Golgi by immuno-fluorescence (Fig. 2b, c), and transmission electron microscopy (TEM) revealed a visibly enlarged intra-cisternal space, a general dis-organization rendering Golgi polarization difficult to determine, and a high level of vesiculation (Supplementary Fig. 2a). In addition, mTORC1 kinase activity as evaluated by the phosphorylation status of its target RPS6K was decreased by ~30%, and consequently autophagy flux was slightly increased, as evidenced by an increase of the lipidated autophagosome marker GABARAPL1, which was stabilized by ConA (Supplementary Fig. 2b, c)[40,41]. SCYL1 interaction with the TGN golgin GORAB is necessary for COPI-coat association to the TGN[36]. Consistently, *SCYL1* KO cells showed a profound spreading of COPI-coat vesicles detected via the COPG1 subunit, but no difference in GORAB localization (Supplementary Fig. 2d, e), confirming the role for SCYL1 as a scaffolding factor promoting the interaction between TGN-resident GORAB and the COPI-coat. Thus, we conclude that our *SCYL1* KO cell lines recapitulate the previously reported phenotypes and can be used to study the role of SCYL1.

To get more insights into the physiological relevance of SCYL1, we performed expression proteomics of *SCYL1* KO and wild-type (WT) MCF-7 cells using stable isotope labeling by amino acids in cell culture (SILAC)-based quantitative mass spectrometry (MS) (Fig. 2d, e). 81 proteins were significantly dysregulated in *SCYL1* KO compared to WT cells in two biological replicates (Fig. 2d; significance A, $p \leq 0.05$, BH-corrected, Supplementary Data 1). Gene ontology (GO) analysis of down-regulated proteins in *SCYL1* KO cells highlighted perturbation of multiple metabolic processes, including cellular respiration, and lipid, amino acid, and nucleotide biosynthesis (Fig. 2e and Supplementary Data 1), indicating that *SCYL1* KO cells have an altered metabolism. This was previously reported for *SCYL1* RNAi cells and correlated with our observations of reduced *SCYL1* KO cell growth[37]. Interestingly, upre-gulated proteins in *SCYL1* KO cells localized to lysosomes and covered processes related to cellular differentiation, intracellular vesicle trafficking and secretion (Fig. 2e and Supplementary Data 1). These data suggested that intracellular transport routes and secretion may change in *SCYL1* KO cells, possibly due to reorganization of endosomal trafficking or maturation. To test this, the subcellular localizations of RAB5, RAB7, and LAMP2 were evaluated in WT and *SCYL1* KO cells. Strikingly, localization of endosomal makers RAB5 and RAB7 was visibly affected by loss of SCYL1; most particularly, RAB5 distribution beneath the plasma membrane in WT cells was lost in *SCYL1* KO cells, while both RAB5 and RAB7 concentrated more prominently to a peri-nuclear location (Fig. 2f, h). Similarly, lysosome subcellular localization changed in *SCYL1* KO cells and lysosomes became intensely focused in large peri-nuclear patches, indicative of a collapsed lysosomal network (Fig. 2g, h). 3D-measurements of the shortest distances to the nuclear surface of RAB5, RAB7 and LAMP2 foci confirmed these observations, with significant differences for RAB7 and LAMP2 distances to nuclei (Fig. 2h). The observed increase in LAMP2 staining in *SCYL1* KO cells corroborated our proteomic analysis showing an increase of lysosome-associated proteins in these mutants (Fig. 2e). Thus, we infer that SCYL1 is involved in the intracellular distribution of the endolysosomal compartment.

## Loss of SCYL1 enhances extracellular vesicle secretion

Endosome-to-lysosome maturation can regulate the balance between lysosomal degradation and extracellular vesicle (EV) secretion[42]. To test whether the redistribution of endosomes and lysosomes observed in *SCYL1*-deficient cells led to changes in protein secretion,

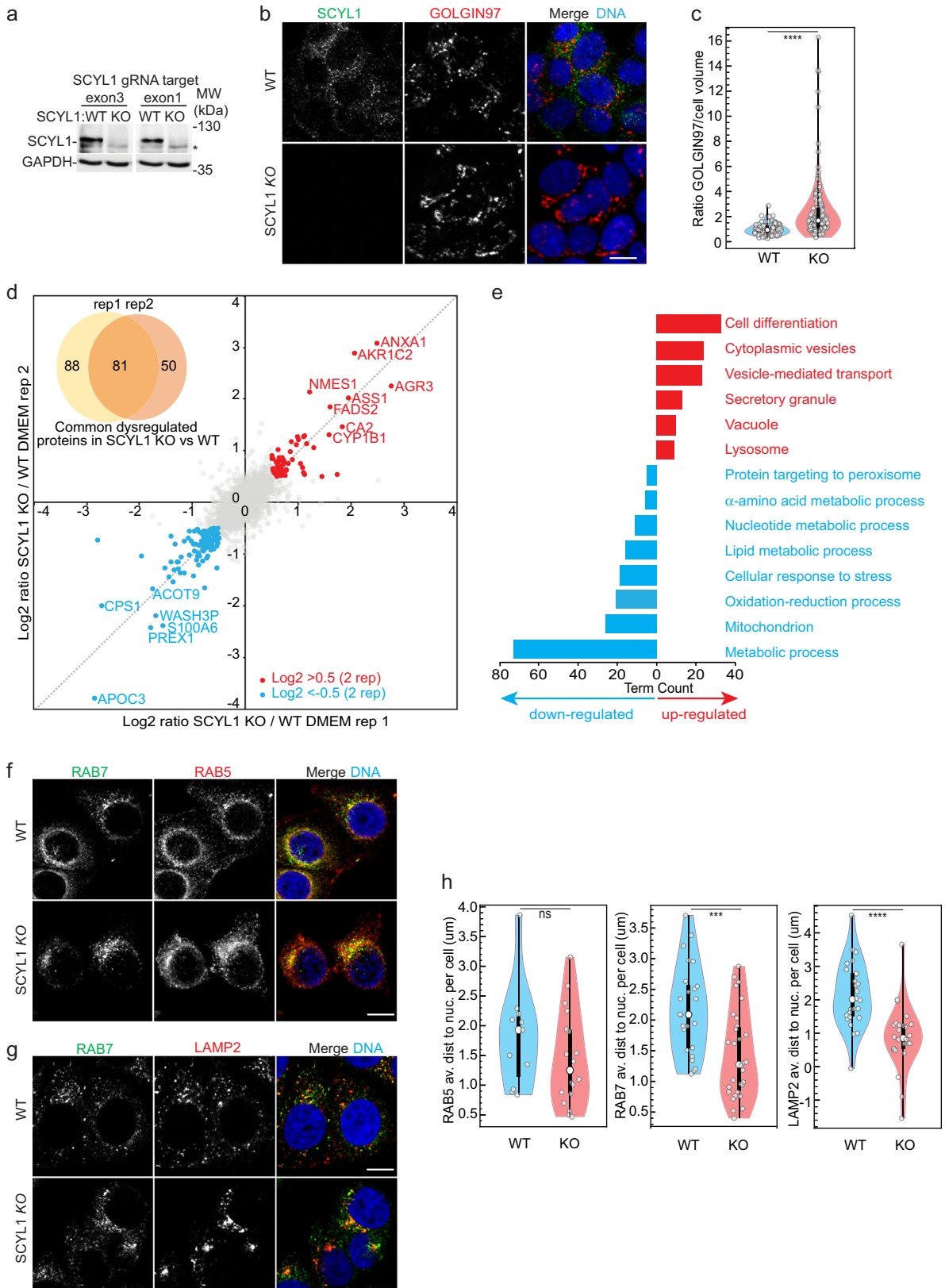

we analyzed the protein content in ultrafiltered (UF, 100 kDa cut-off) conditioned medium (CM) of SILAC-labeled WT and *SCYL1* KO cells by proteomic profiling (Fig. 3a and Supplementary Data 2). Numerous proteins were enriched in CM of *SCYL1* KO cells, indicating increased protein and/or vesicle secretion. Strikingly, both *SCYL1* KO clones presented a strong enrichment in CD9, CD81, CD63, CD151, CD99 and

CD82, six tetraspanins that are established EV marker proteins (Fig. 3a)[43]. Using ultracentrifugation followed by western blot, we confirmed that the CM of *SCYL1* KO cells contained high levels of CD9 (Fig. 3b, c). Other exosome components including ESCRT-III subunits, integrins, ferritin, complement subunits, and syntenin/syndecan-associated proteins were also strongly secreted by *SCYL1* KO cells

**Fig. 2 | SCYL1 regulates Golgi structure and endolysosomal positioning.**
**a** Generation of *SCYL1* KO cells. Two *SCYL1* CRISPR/cas9 KO clones targeting exon 3 (KO1) or exon 1 (KO2) of the *SCYL1* gene ORF, were generated in MCF-7 cells. GAPDH was used as loading control. Equal protein amounts were loaded. One representative western blot out of $n = 3$ biological replicates is shown. *: nonspecific background signal. **b** *SCYL1* KO cells display an enlarged Golgi. Representative fluorescence micrographs from WT and *SCYL1* KO MCF-7 cells (clone KO1) are shown. Scale bar, 10 μm. **c** Quantification of **b**. The ratio between the Golgi apparatus volume, labeled with GOLGIN-97/GOLGA1 antibodies, and the total cellular volume, detected with the cell body detection tool (Imaris), was computed in individual cells. Violin plots were generated using InstantClue (white dots within violins denote medians; top and bottom of violins represent the 25th and 75th percentiles; thick vertical lines within violins show the interquartile range (IQR), whereas thin vertical lines represent the rest of the distribution extending on the 1.5xIQR rule)[75]. Single-cell values are indicated with white small dots. An unpaired two-tailed Student's *t*-test was used to compare WT and *SCYL1* KO values: $p = 8.05e$-$09$ (****). $n = 3$ biological replicates with at least 36 individual cells measured per replicate. Error bar = SEM. **d** Expression proteomics comparing *SCYL1 KO* and WT MCF-7 cells. Scatter plot showing protein abundance ratios of SILAC-labeled *SCYL1* KO and WT cells of two biological replicates. Normalized ratios were $Log_2$-transformed. Pearson's correlation coefficient $r = 0.74$. Red dots: significantly upregulated proteins ($Log_2 \geq 0.5$ in both replicates) in *SCYL1* KO compared to WT cells; blue dots: significantly down-regulated proteins ($Log_2 \leq 0.5$ in both replicates) in *SCYL1 KO* compared to WT cells. Upper left corner: Venn diagram quantifying common proteins significantly dysregulated in both *SCYL1 KO* replicates compared to WT (*t*-test two-sided, FDR = 0.05, BH-corrected). Raw data are provided in Supplementary Data. **e** Proteins involved in vesicular transport and secretion are increased in *SCYL1* KO cells. GO term enrichment of upregulated (red) and down-regulated (blue) proteins in the *SCYL1 KO* proteome, as highlighted in **d**. A complete list of all significant GO terms is displayed in Supplementary Data 1. **f, g** Loss of *SCYL1* leads to a perturbation of early and late endosome (**f**) and lysosome (**g**) distribution. Fluorescence micrograph from WT and *SCYL1* KO MCF-7 cells in normal growth conditions (DMEM). Representative pictures of $n = 3$ replicates are shown. Scale bar = 10 μm (**h**). Quantification of **f, g**. The distance from nuclear envelope to RAB5, RAB7 and LAMP2 foci was computed and averaged per individual cell. All results for WT and *SCYL1* KO cells were assembled in a violin plot (see panel **c** for details). Single-cell values are indicated with white small dots. An unpaired two-tailed Student's *t*-test was used to compare WT and *SCYL1* KO populations; RAB5: $p = 0.38$ (ns); RAB7: $p = 0.00044$ (***); LAMP2: $p = 1.88E$-$05$ (****). 10 individual cells were individually measured within three technical replicates per two biological replicates ($n = 6$). Error bar, SEM. All source data are provided as Source Data File.

(Fig. 3a) (listed in www.exocarta.org, reviewed in refs. 44,45, see also refs. 46,47). We categorized the significantly enriched proteins in *SCYL1* KO CM according to the MISEV2018 guidelines for global characterization of EVs[48], and identified multiple proteins belonging to categories 1 and 2, indicating the presence of EVs (Supplementary Data 2). Altogether, these data reveal that protein secretion, and potentially secretion of EVs, is significantly enhanced in *SCYL1* KO cells.

To test whether the increase in secreted proteins did indeed correspond to increased release of EVs, we examined ultracentrifuged CM from WT and *SCYL1* KO cells by scanning electron microscopy (SEM) and nanoparticle tracking (NTA) (Fig. 3d, e and Supplementary Fig. 3a, b). WT MCF-7 cells released heterogenous vesicle populations composed of vesicles with a diameter of <200 nm and of <400 nm (Fig. 3d, e and Supplementary Fig. 3a, b), probably corresponding to a mixture of small EVs (sEVs) and medium to large EVs, respectively (nomenclature according to MISEV2018 guidelines[48]). In contrast, *SCYL1* KO cells released a significant higher number of more homogenous vesicles with a diameter of ≤200 nm, corresponding to sEVs (Fig. 3d, e and Supplementary Fig. 3a, b). We therefore concluded that SCYL1 directly or indirectly negatively regulates EV secretion in growing conditions.

## Loss of *GORAB* does not lead to increased EV release
SCYL1 recruitment to the TGN is ensured by GORAB, a major SCYL1 interactor[36]. Therefore, loss of *GORAB* expression should lead to SCYL1 detachment from the TGN. If mTORC1-mediated SCYL1 dissociation from the TGN is critical for the observed EV release, *GORAB* KO cells should recapitulate the increased EV secretion phenotype of *SCYL1* KO cells. To test this, we generated CRISPR/Cas9 *GORAB* KO cells and monitored SCYL1 expression and localization in absence of GORAB by western blotting and immunofluorescence, respectively (Fig. 3f and Supplementary Fig. 3c–e). GORAB depletion had no major effect on SCYL1 levels or mTORC1 activity in soluble protein extracts, but a slight stabilization of SCYL1 and its interactor COPB1 was visible under mTORC1 inhibiting conditions (Fig. 3f). In untreated *GORAB* KO cells, we observed a rise in peripheral SCYL1-positive puncta compared to WT (Supplementary Fig. 3c, d), confirming the redistribution of SCYL1 from the TGN in absence of GORAB. However, some peri-nuclear SCYL1 staining remained, probably due to GORAB-independent recruitment of SCYL1 to cis-Golgi and ERGIC, as previously observed (Supplementary Fig. 3c, d)[36]. Rapamycin treatment had no effect on the peripheral SCYL1 pool in *GORAB* KO cells, given that it was already elevated in basal conditions (Supplementary Fig. 3c, d). mTORC1

inhibition and loss of GORAB functions therefore both lead to a dissociation of SCYL1 from the TGN and an increase of peripheral SCYL1 foci.

Next, we studied the effect of *GORAB* KO on protein secretion by proteomic profiling comparing *GORAB* KO and *SCYL1* KO CM to WT CM (Fig. 3g, and Supplementary Data 2). As before, *SCYL1* KO induced pronounced protein secretion, marked by elevated levels of EV proteins CD9, CD81, and ESCRT-III subunits CHMP1-5 (Fig. 3g). Surprisingly, *GORAB* KO and WT cells displayed comparable amounts of CD9, CD81, and ESCRT subunits in their CM (Fig. 3g), suggesting that the loss of *GORAB* and subsequent SCYL1 dissociation from the TGN were not sufficient to provoke EV release. These observations were validated by additional proteomic analyses of CM of *SCYL1* KO and *GORAB* KO compared to WT cells (Fig. 3h), and by respective western blot analyses (Supplementary Fig. 3e). In both cases *GORAB* KO did not lead to increased secretion of EV proteins, and therefore did not phenocopy *SCYL1* KO. Interestingly, the effects of rapamycin on secretion were additive to those caused by *SCYL1* depletion alone, supporting the idea that mTORC1 inhibition has broader consequences on EV secretion than the exclusive loss of *SCYL1* (Supplementary Fig. 3e). These data imply that SCYL1 negatively regulates EV secretion when mTORC1 is active and this does not involve GORAB-mediated recruitment to the Golgi.

## mTORC1 phosphorylates SCYL1 on Ser754
We next turned to phosphoproteomics to study whether mTORC1 regulates SCYL1 function through direct phosphorylation. Thus, we generated a detailed map of SCYL1 phosphorylation sites under growth conditions, i.e. when mTORC1 is active, by using affinity purifications (APs) of exogenously expressed HA-SCYL1 and subsequent LC-MS/MS phosphoproteomic analyses. Five phosphorylation sites (Ser41, Ser558, Ser559, Ser732 and Ser754) were identified, among which Ser754 was the most frequently detected (Fig. 4a). Analyzing the stoichiometry of endogenous SCYL1 Ser754 phosphorylation showed an occupancy of 87% ± 11% under growth conditions ($n = 3$)[49], suggesting that this posttranslational modification is a frequent event. Consistently, this specific phosphorylation site was detected in 89 different high throughput MS/MS studies, according to the PhositePlus database (Fig. 4a)[50]. Moreover, a recent global functional phosphoproteomic study aiming to identify phosphosites of structural, regulatory, or evolutionary relevance also attributed SCYL1 Ser754 a significant functional score (Fig. 4a)[51]. Altogether, these observations strongly underscore that SCYL1 is phosphorylated at Ser754 in growth conditions.

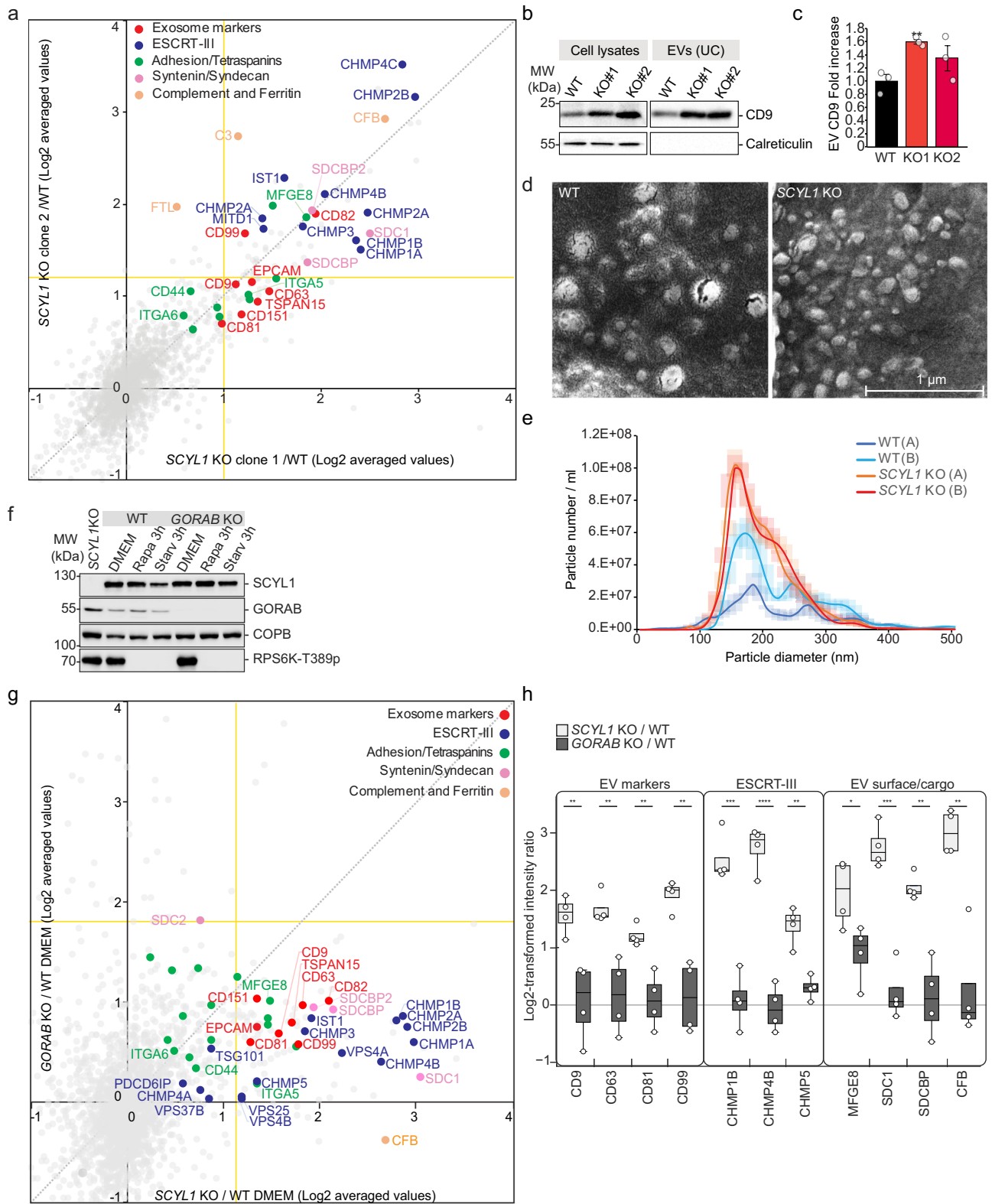

To investigate whether SCYL1 Ser754 is indeed an mTORC1 target, we performed kinetic phosphoproteomic analyses and found that Ser754 is rapidly dephosphorylated upon nutrient starvation or rapamycin treatment (Fig. 4b)[23], fulfilling the requirements for a bona fide mTORC1 target site. To establish whether Ser754 is a direct mTORC1 phosphosite, we performed in vitro kinase assays. Briefly, immunoprecipitated HA-SCYL1 was dephosphorylated by λ-phosphatase to remove endogenous phosphate groups. Purified HA-SCYL1 was then

incubated with either the kinase subunit MTOR purified from HEK293T cells (see Methods for details), or entire mTORC1 complex (consisting of MTOR, LST8 and RPTOR) immunoprecipitated from HeLa cells overexpressing FLAG-MTOR. These experiments were carried out in both the absence and the presence of wortmannin, a well-characterized phosphatidylinositol-3-kinase inhibitor that potently blocks mTORC1[52,53]. To discriminate phosphatase resistant sites from sites phosphorylated in vitro, γ-$^{18}O_4$-labeled ATP was used in the

**Fig. 3 | Loss of *SCYL1* does lead to a marked increase in EV release independent of *GORAB*. a** Proteomic profiling of CM of SILAC-labeled *SCYL1* KO vs. WT MCF-7 cells. Two independent *SCYL1 KO* clones were tested. Scatter plot shows averages of normalized and $log_2$-transformed SILAC ratios. EV components are highlighted. $n = 8$ biological replicates for clone 1, $n = 2$ for clone 2. Yellow lines indicate the significance threshold selected via a two-sided, unpaired *t*-test, threshold $p \leq 0.05$. Raw data are available in Supplementary Data 2. **b** Increased levels of CD9 in CM of SCYL1 KO cells. Western blot analysis of WT and two *SCYL1* KO MCF-7 clones. Cell lysates and ultracentrifuged (UC) extracellular vesicles (EVs) are analyzed. Equal cell numbers were used to normalize protein content. ER marker calreticulin is used as intracellular control. One representative experiment, out of three biological replicates, is shown. **c** Quantification of **b**, average of $n = 3$ biological replicates. Error bars = SEM, unpaired two-tailed Student's *t*-test, KO1: $p = 0.0046$ (**); KO2: $p = 0.18$ (ns). **d** *SCYL1* KO cells secrete vesicles within the size range of exosomes. Representative SEM images of ultracentrifuged WT and *SCYL1* KO MCF-7 EV populations. EVs were purified from equal amounts of seeded cells. Scale bar = 1 μ m. One representative experiment, out of n = 2 independent replicates, is shown (see also Supplementary Fig. 4). **e** NTA confirms that *SCYl1* KO cells secrete more EVs. NTA of ultracentrifuged EVs from equal numbers of WT and *SCYL1 KO* MCF-7 cells, in biological duplicates. For each biological replicate, the averages of six technical replicates in two independent movies were calculated. Shaded areas represent the SEM for each data point. **f** *GORAB* KO MCF-7 cells. KO cells were generated by CRISPR/Cas9 and their protein levels monitored by western blotting, in normal (DMEM) or mTORC1-inhibited conditions (Rapa 100 nM or starvation in HBSS for 3 h). Equal protein amounts were loaded. One representative western blot, out of $n = 3$ biological replicates, is shown. **g** Protein profiling in CM of SILAC-labeled *SCYL1* KO vs. *GORAB* KO MCF-7 cells. Scatter plot showing normalized, averaged and $Log_2$-transformed SILAC ratios of CM of *SCYL1* KO/WT and *GORAB* KO/WT comparisons. EV components are highlighted. $n = 4$ biological replicates for *SCYL1* KO/WT, $n = 2$ for *GORAB* KO/WT. Yellow lines indicate the significance threshold selected via a two-sided, unpaired *t*-test, threshold $p \leq 0.05$. Raw data are available in Supplementary Data 2. **h** *GORAB* KO cells do not phenocopy *SCYL1* KO cells. Box plots showing individual EV components and cargoes enrichment in CM of *SCYL1* KO/WT and *GORAB* KO/WT cells, as shown in **g**. $n = 4$ biological replicates per condition, indicated by white dots. Gray line marks the 0 value, indicating no difference. Center lines within the box plots indicate median values; box limits indicate the upper and lower quartiles, whereas upper and lower whiskers indicate the largest and smallest values within the 1.5x IQR. An unpaired two-tailed Student's *t*-test was used to compare *SCYL1* KO/WT and *GORAB* KO/WT values; CD9: $p = 0.0067$(**); CD63: $p = 0.0050$(**); CD81: $p = 0.0045$(**); CD99: $p = 0.0019$(**); CHMP1B: $p = 0.00025$(***); CHMP4B: $p = 5.71e\text{-}05$; CMHP5: $p = 0.0014$(**); MFGE8: $p = 0.034$(*); SDC1: $p = 0.00017$(***);S DCBP: $p = 0.0016$(**); CFB: $p = 0.0017$(**). All source data are provided as Source Data file.

reaction. Phosphorylated peptides marked by heavy phosphate were quantified by LC-MS/MS. Ser754 was the only site identified reproducibly as an mTORC1-specific phosphosite sensitive to wortmannin (Fig. 4c and Supplementary Data 3).

To confirm that SCYL1 Ser754 is phosphorylated by mTORC1, we generated a phosphoSer754-specific antibody, which displayed ~80% specificity to the phosphorylated form of SCYL1 (Supplementary Fig. 4a–c). Time-course western blot analyses recapitulating the phosphoproteomic studies of Fig. 4b confirmed that SCYL1 Ser754 was rapidly dephosphorylated upon mTORC1 inhibition by rapamycin or nutrient starvation (Fig. 4d, e), although the decrease was less pronounced, probably due to the partial cross-reactivity of the antibody with the non-phosphorylated version of the protein. Treatment with Torin1, an mTORC1/2-specific inhibitor[54], also led to a marked and rapid decrease in SCYL1 Ser754 phosphorylation with similar kinetics (Fig. 4d, e). Compared to RPS6K Thr389 phosphorylation, mTORC1 inhibition led to a less dramatic reduction of SCYL1 phosphorylation, indicating distinct regulation of both downstream targets, SCYL1 being potentially regulated by additional kinases. To analyze the kinetics of SCYL1 Ser754 de novo phosphorylation by mTORC1 in growth conditions, MCF-7 WT cells were starved in HBSS and released to complete growth medium supplemented with cycloheximide, which boosts mTORC1 activation by increasing the intracellular amino acid pool[55]. Similar to RPS6K re-phosphorylation kinetics, SCYL1 Ser754 phosphorylation rapidly increased (Fig. 4f, g), further supporting that this phosphosite is directly dependent on mTORC1 activity. Taken together, our results demonstrate that SCYL1 Ser754 is directly phosphorylated by mTORC1, and that pharmacological and physiological mTORC1 inactivation leads to its rapid, yet partial, dephosphorylation.

## mTORC1 phosphorylation of SCYL1 controls its localization

To understand the specific role of SCYL1 phosphorylation by mTORC1, inducible *HA-SCYL1(WT)*, non-phosphorylatable *HA-SCYL1(S754A)*, and phospho-mimicking *HA-SCYL1(S754E)* transgenes were expressed in *SCYL1* KO cells. Expression levels of the three HA-tagged constructs were slightly higher than those of the endogenous protein, equal expression being reached after 24 h doxycycline induction and increasing up to 48 h (Supplementary Fig. 5a). As loss of *SCYL1* leads to Golgi enlargement, we first checked whether HA-SCYL1(WT) is fully functional by examining the Golgi architecture, and whether SCYL1 phosphorylation plays a role in this function. TEM of MCF-7 *SCYL1* KO cells re-expressing HA-SCYL1(WT), HA-SCYL1(S754A), or HA-SCYL1(S754E) for 24 h revealed that HA-SCYL1(WT) rescued efficiently Golgi architecture defects of *SCYL1* KO cells, while HA-SCYL1(S754A) mutant did not complement the enlarged cisterns and disorganized Golgi ultrastructure (Fig. 5a and Supplementary Fig. 5b). HA-SCYL1(S754E) expression had an intermediate effect, with most Golgi structures having a WT appearance, while a few "diffuse" structures persisted (Fig. 5a and Supplementary Fig. 5b). Similarly, measurements of GOLGIN-97/GOLGA1 distribution by immunofluorescence as proxy of TGN size in *SCYL1* KO cells re-expressing HA-SCYL1(WT), HA-SCYL1(S754A), or HA-SCYL1(S754E) mutants demonstrated that SCYL1(WT), as well as phospho-mimicking SCYL1(S754E), rescued TGN structure proportionally to the amount of re-expressed fusion protein (Fig. 5b, c). In contrast, phospho-null HA-SCYL1(S754A) was unable to complement the TGN-associated phenotype of *SCYL1* KO cells even after 48 h of induction (Fig. 5b, c). These results show that mTORC1-mediated phosphorylation on Ser754 is essential for the function of SCYL1 in Golgi architecture maintenance.

The subcellular localization of HA-SCYL1(WT), HA-SCYL1(S754A), or HA-SCYL1(S754E) was also analyzed by immunofluorescence (Fig. 5b). While HA-SCYL1(WT) and HA-SCYL1(S754E) localization was mostly peri-nuclear, consistent with a predominant Golgi/ERGIC localization, HA-SCYL1(S754A) exhibited a widespread cytosolic localization in puncta, including in regions below the PM (Fig. 5b). This latter observation demonstrates that Ser754 phosphorylation is important to maintain SCYL1 at the Golgi and ERGIC. Notably, HA-SCYL1(S754E) did not display a full WT localization and appeared more concentrated in the peri-nuclear area (Fig. 5b), which could indicate that dephosphorylation is essential for redistribution of SCYL1 from the Golgi/ERGIC region to endosomes.

## SCYL1 phosphorylation regulates EV secretion

As *SCYL1* KO cells showed a peri-nuclear concentration of endosomes and lysosomes (Fig. 2f–h), we tested whether SCYL1 Ser754 phosphorylation also affects the subcellular distribution of these compartments. To that aim, RAB5, RAB7 and LAMP2 localization was analyzed in *SCYL1* KO cells re-expressing HA-SCYL1(WT), HA-SCYL1(S754A), and HA-SCYL1(S754E) variants (Fig. 6a–c). Upon 24 h of doxycycline induction, HA-SCYL1(WT) and to a lesser extent also HA-SCYL1(S754E) expressing cells exhibited a punctate RAB5 and RAB7 distribution throughout the cytoplasm similar to the one observed in WT cells (Fig. 6a–c and Fig. 1i, j). RAB5 and RAB7 foci distance to the nuclear surface significantly increased in both cell lines, supporting the rescue of KO phenotypes (Fig. 6c). LAMP2 responded slower and its distribution was significantly rescued to WT appearance after 36 h of

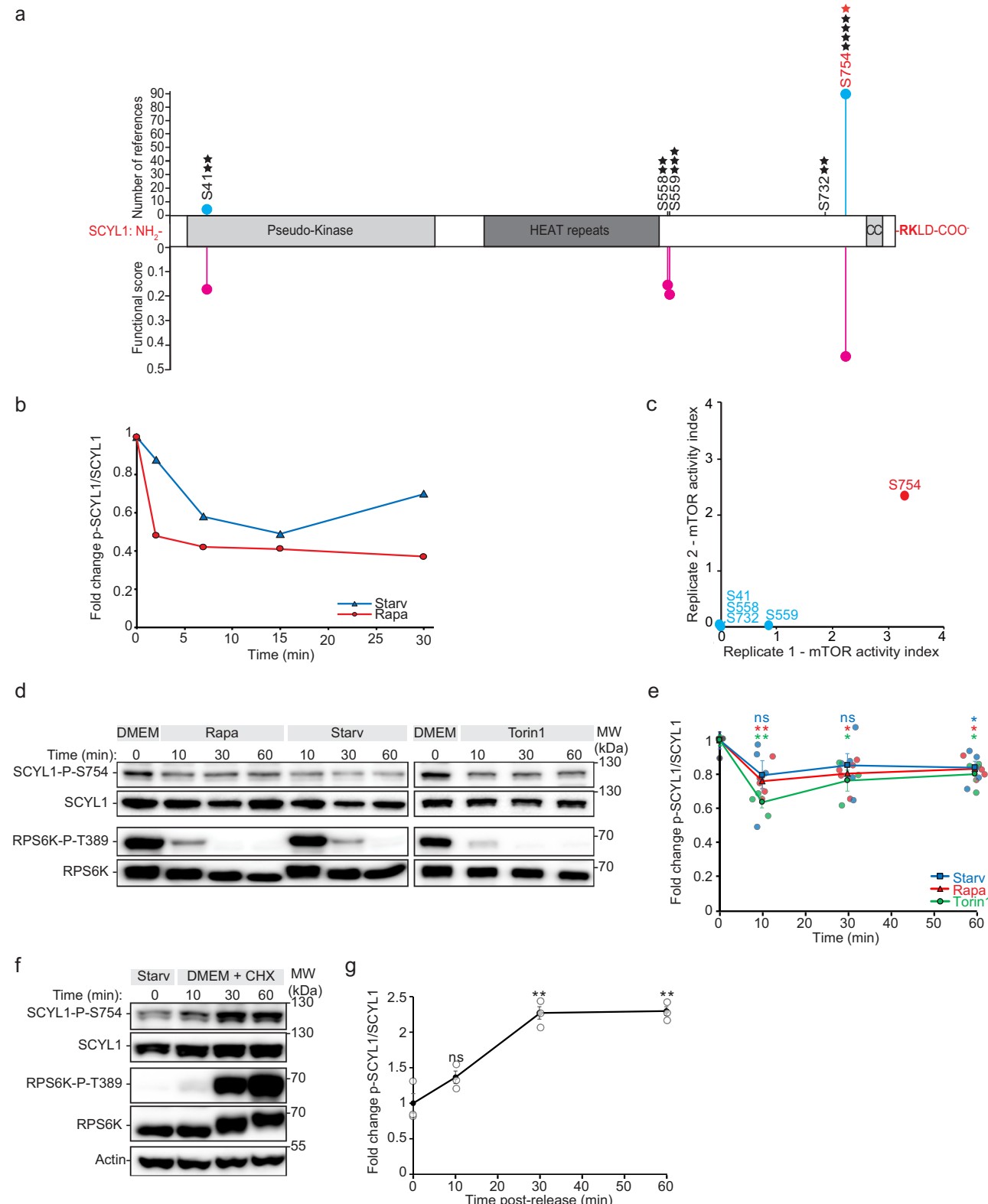

HA-SCYL1(WT) and HA-SCYL1(S754E) induction, suggesting that its recovery was chronologically successive to the repositioning of early and late endosomes (Fig. 6b, c). In contrast, the HA-SCYL1(S754A) mutant was unable to rescue RAB5, RAB7 and LAMP2 localizations, displaying a peri-nuclear concentration of endosomes and lysosomes similar to the one observed in *SCYL1* KO cells (Fig. 6a, c).

Given that mTORC1 was also shown to be an inhibitor of exosome secretion[56], and that *SCYL1* KO cells display increased EV secretion, we

postulated that SCYL1 phosphorylation at Ser754 could modulate EV release. The effect of re-expressing HA-SCYL1(WT), HA-SCYL1(S754A), and HA-SCYL1(S754E) variants on *SCYL1* KO-induced EV secretion was monitored by western blotting of ultracentrifuged CM (Fig. 6d, e). Upon 24 h of re-expression, HA-SCYL1(WT) and HA-SCYL1(S754E) strongly reduced *SCYL1* KO-induced EV release as indicated by lower CD9 and CD81 protein levels in CM, whereas HA-SCYL1(S754A) only partially rescued the KO phenotype and showed significantly higher

**Fig. 4 | SCYL1 Ser754 is a novel mTORC1-regulated phosphosite. a** Schematic representation of SCYL1 protein structure. Known conserved domains and phosphorylation sites identified by MS are shown. CC = Coiled-coil domain. Black stars: phosphorylation events detected in independent AP-MS experiments; Red star: Ser754, identified reproducibly as regulated in in vitro mTOR kinase assays. Blue lollipop chart: number of citations referenced in Phosphosite Plus®[50] for indicated phosphosites. Pink lollipop chart: functional score for indicated phosphosites, as published in ref. [51]. **b** SCYL1 Ser754 is rapidly dephosphorylated upon mTORC1 inhibition. Phosphoproteomic time-course of SCYL1 Ser754 phosphorylation levels in SILAC-labeled MCF-7 WT cells upon HBSS starvation or 100 nM rapamycin treatment. Average values of $n = 2$ biological replicates. **c** In vitro mTOR kinase assay qualifies SCYL1 Ser754 as direct mTOR target site. Purified mTOR was incubated with immunoprecipitated HA-SCYL1 in absence or presence of wortmannin (12 μM, 30 min). Two biological replicates were analyzed by LC-MS/MS. Results are displayed as a ratio of phosphosite intensities without/with wortmannin (Supplementary Data 3). **d** Time-course of SCYL1 Ser754 phosphorylation upon mTORC1 inhibition, detected by a phosphosite-specific antibody. WT MCF-7 cells were left untreated or treated with 100 nM rapamycin, starvation (HBSS), or 250 μM Torin1, for the indicated times. One representative time-course experiment out of three biological replicates is shown. Equal protein amounts were loaded. **e** Quantification of **d**. Phospho-Ser754-SCYL1 levels were normalized to total SCYL1 levels. Average of $n = 4$ biological replicates for DMEM/Rapa and DMEM/Starv measurements, and $n = 3$ biological replicates for DMEM/Torin1 measurements. Red line: rapamycin/ DMEM 0 min; blue line: starvation/DMEM 0 min; green lines: Torin1/DMEM 0 min. All single values are indicated by colored dots. *p*-values: An unpaired two-tailed Student's *t*-test was used to compare the indicated time points with the 0 min reference; Rapa10′/DMEM $p = 0.0068$(**); Rapa 30′/DMEM $p = 0.017$(*); Rapa 60′/ DMEM $p = 0.018$(*); HBSS 60′/DMEM $p = 0.038$(*); Torin10′/DMEM $p = 0.0029$(**); Torin30′/DMEM $p = 0.049$(*); Torin60′/DMEM $p = 0.043$(*). Error bars, SEM. **f** SCYL1 Ser754 phosphorylation responds to mTORC1 reactivation. WT MCF-7 cells were starved in HBSS for 4 h, then released in complete DMEM supplemented with 50 μg/ ml cycloheximide to increase the intracellular amino acid concentration for the indicated times. One representative time-course experiment out of three biological replicates is shown. Equal protein amounts were loaded. **g** Quantification of **f**. Phospho-Ser754-SCYL1 levels were normalized to total SCYL1 levels. Average of $n = 3$ biological replicates. *p*-values: An unpaired two-tailed Student's *t*-test was used to compare the indicated time points with the starved cells reference time point (before release); 30′: $p = 0.0028$(**); 60′: $p = 0.0018$(**). Error bars, SEM. All source data are provided as Source Data file.

---

levels of CD9 and CD81 EV marker proteins than HA-SCYL1(WT) re-expressing cells (Fig. 6d, e). Thus, SCYL1 phosphorylation on Ser754 by mTORC1 appears to be directly responsible for inhibiting EV release in growth conditions.

## Phosphorylation of SCYL1 influences its protein neighborhood

To substantiate that mTORC1 phosphorylation leads to a redistribution of SCYL1, we performed proximity labeling-MS to identify neighboring proteins of SCYL1(S754A) and SCYL1(S754E). We inducibly expressed these proteins fused to the biotin ligase BirA in *SCYL1* KO cells, BirA-only expressing cells serving as negative control, biotinylated proximal proteins, and compared streptavidin bead-based enrichments by SILAC-based quantitative proteomics (Fig. 7a). Point mutations had no influence on GORAB or COPI/Coatomer complex members (Fig. 7b, d), consistently with our finding that SCYL1 regulation by mTORC1 does not depend on GORAB (Fig. 3g, h). However, differences were identified. Proteins enriched stronger in SCYL1(S754A)-BirA principally carried GO/KEGG terms related to early endosome, endosome to Golgi retrograde transport, and actin cytoskeleton regulation (Fig. 7b, c and Supplementary Data 4), further supporting our finding that dephosphorylated SCYL1 relocalizes to endosomes, and proposing that the cytoskeleton supports this migration. In contrast, proteins enriched stronger in SCYL1(S754E)-BirA belonged to Golgi, ER and mitochondria, also agreeing with our observation that phosphorylation retains SCYL1 in its original membranous compartments (Fig. 7b, c). Single candidate analyses of these categories confirmed this global trend highlighting significant differences (Fig. 7d). Our interaction assays therefore confirm that mTORC1-dependent Ser754 phosphorylation leads to a change in SCYL1's subcellular distribution and regulates its associations with Golgi and endosomal membranes.

## Discussion

mTORC1 is a master regulator of cell homeostasis. Under growth conditions, the predominant site of mTORC1 localization are lysosomal/vacuolar membranes via interaction with the EGO complex in yeast[1,57], or Rag GTPases and the Ragulator complex in mammalian cells[10,58]. Lysosomal localization supports sensing of amino acids, which in turn allows mTORC1 to adjust cell growth, i.e. anabolism, to nutrient availability. Recently, several additional subcellular localizations of mTORC1 have been described. In mammalian cells, a Golgi pool of mTORC1 was identified that preferentially phosphorylated 4E-BP1 in comparison to RPS6K[19,20]. An mTORC1 pool on so-called "signaling endosomes" has been characterized[15], which in yeast appears to be involved in the negative regulation of micro- and macroautophagy via phosphorylation of Vps27 and Atg13, respectively[12]. Thus, distinct mTORC1 regulatory functions seem to be intimately linked to discrete subcellular localizations. With respect to secretion, mTORC1 was shown to regulate unconventional protein secretion via GRASP55[21]. In addition, mTORC1 hyperactivity in TSC2-/- mice was shown to negatively regulate Rab27-dependent exosome release, leading to intracellular accumulation of exosomal precursor proteins;[56] however, the underlying molecular mechanisms regulating EV secretion remained elusive. Here, we characterize SCYL1 as a Golgi-associated mTORC1 target protein, and highlight how mTORC1 regulates Golgi architecture, subcellular distribution of the endolysosomal compartment, and EV release via phosphorylation and redistribution of the scaffold protein SCYL1.

Approximately 90% of the SCYL1 pool appear to be phosphorylated on Ser754 under growth conditions, which is in line with the frequent identification of this phosphorylation site in global, unbiased phosphoproteomic analyses. This high phosphosite occupancy indicates an important homeostatic role, and we could generate experimental evidence that Ser754 phosphorylation is key for SCYL1 function, as re-expression of a non-phosphorylatable alanine mutant could not rescue the phenotypes of *SCYL1* KO cells. The observation that even in growth inhibiting conditions, i.e., inactivation of mTORC1, ~45% of the total SCYL1 pool appears to be still phosphorylated might indicate that next to mTORC1 additional kinases can phosphorylate this site, or that this site is relatively resistant to phosphatase activity. With respect to a potential scaffold function of SCYL1 at different cellular compartments both hypotheses are interesting and should be addressed in future studies.

The detailed molecular functions of SCYL1 seem to be complex and not limited to its function as a TGN scaffold for COPI-coat recruitment via binding to the golgin GORAB[36]. *GORAB* knockout, which leads to a dissociation of SCYL1 from the TGN, did not phenocopy the increase in EV release observed in *SCYL1* KO- or *SCYL1^S754A^*-expressing cells. Thus, it appears that the scaffold function of SCYL1 is more general, which is supported by the observation that SCYL1 is present on several organelles, including Golgi, ERGIC, early and late endosomes. In addition, we identified subunits of the COPII-coat and the clathrin adaptor AP2 complex in the vicinity of SCYL1, suggesting that SCYL1 could also be present at the plasma membrane. COPI and COPII coats have a common mode of assembly as clathrin-like cages, and some of their subunits share strikingly similar structures and functions (reviewed in ref. [59]). This raises the possibility that SCYL1 scaffolding capacities extend to all coated vesicles.

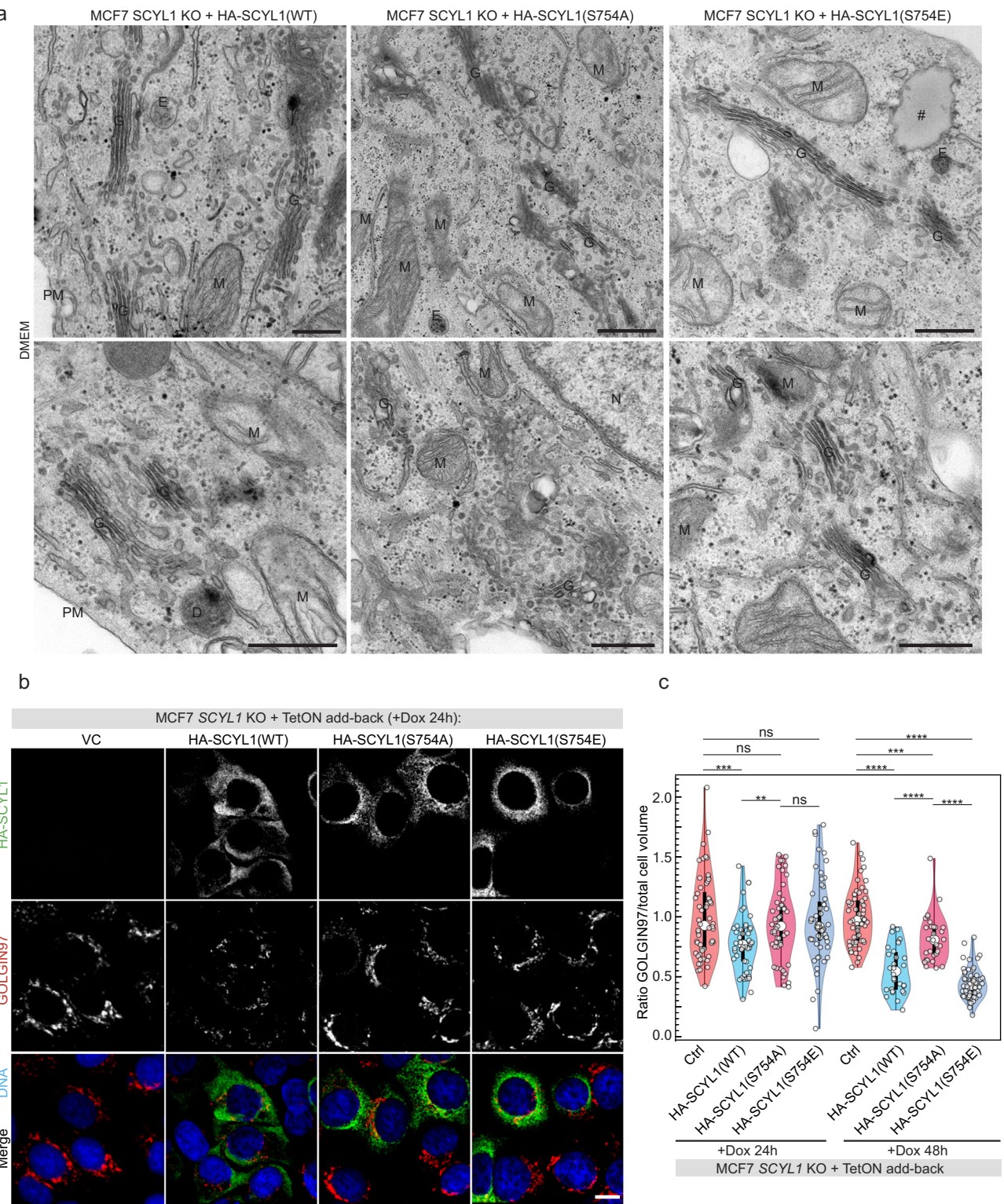

Interestingly, we detected subtle compartment differences between the SCYL1(S754A) and SCYL1(S754E) mutants by proximity labeling MS/MS, which are supported by our immunofluorescence analyses. This might indicate that SCYL1 interaction changes are very dynamic, or that additional phosphorylation sites contribute to SCYL1 regulation. The latter is to be expected, since our phosphoproteomic analyses identified additional, though mTORC1-independent, SCYL1 phophosites. However, SCYL1 might also directly interact with different lipid classes that could guide its subcellular localization without interfering with its protein-protein interactions.

Our proximity labeling results do not allow us to discriminate whether COPI and COPII vesicles are scaffolded by the same SCYL1 oligomers, or by different pools. They do however indicate that these interactions are strong and unperturbed by mTORC1 inhibition. Noticeably, COPI-coated vesicle trafficking is not only important for retrograde transport of ER-resident proteins, it is also essential to promote proper endosome maturation and autophagy[60], as loss of COPI-coat function leads to perturbations in early endosome trafficking and maturation, and autophagy flux[60].

**Fig. 5 | Ser754 phosphorylation modulates SCYL1 functions in Golgi structure maintenance. a** Golgi structure depends on SCYL1 Ser754 phosphorylation. Two representative TEM pictures of MCF-7 *SCYL1* KO cells re-expressing either HA-SCYL1(WT), HA-SCYL(S754A) or HA-SCYL1(S754E) upon 24 h doxycycline induction are shown. D, degradative compartment; E, endosome; G, Golgi apparatus; M, mitochondria; N, nucleus; PM, plasma membrane; #, lipid droplets. Scale bar: 500 nm. Additional pictures are available in Supplementary Fig. 5. **b** SCYL1 Ser754 phosphorylation maintains Golgi architecture. Contrary to HA-SCYL1(WT) and HA-SCYL1(S754E) proteins, which localize closer to the Golgi and around the nucleus, HA-SCYL(S754A) phospho-null localizes to the cell periphery. The enlarged Golgi observed in *SCYL1* KO cells is rescued by HA-SCYL1(WT) and HA-SCYL1(S754E), but not by HA-SCYL1(S754A). Representative fluorescent micrograph of *SCYL1* KO MCF-7 cells re-expressing HA-SCYL1(WT), HA-SCYL1(S754A) or HA-SCYL1(S754E) upon 24 h doxycycline induction. *N* = 3 biological replicates. **c** Quantification of **b**. The ratio between the Golgi volume, labeled with GOLGIN-97/GOLGA1 antibodies, and the total cellular volume, detected with the cell body detection tool (Imaris), was computed in individual cells. All results for *SCYL1* KO MCF-7 cells re-expressing HA-SCYL1(WT), HA-SCYL1(S754A), or HA-SCYL1(S754E) for 24 h or 48 h were assembled in a violin plot (white dots within violins denote medians; top and bottom of violins represent the 25th and 75th percentiles; thick vertical lines within violins show the IQR, whereas thin vertical lines represent the rest of the distribution extending the 1.5xIQR). Single-cell values are indicated with small white dots. An unpaired two-tailed Student's *t*-test was used for comparison: 24 h + WT/VC: $p = 0.00037$(***) and /S754A: $p = 0.0036$(**); 48 h + WT/VC: $p = 9.72e\text{-}14$(****) and /S754A: $p = 2.98e\text{-}06$(****); 48 h + S754A/VC: $p = 0.00066$(***); 48 h + S754E/VC: $p = 2.21e\text{-}29$(****) and /S754A: $p = 5.37e\text{-}17$(****). ≥30 cells were measured amongst technical triplicates of two biological replicates ($n = 6$). Error bars = SEM. Source data are provided as Source Data file.

To our surprise, SCYL1 interaction with RAB6-interacting GORAB was neither perturbed by mTORC1 inhibitory treatments, nor important for SCYL1 functions in organelle repositioning and EV release. Although *GORAB* loss of function generated visible SCYL1 dissociation from the Golgi, it did not lead to increased EV secretion. Loss of *GORAB* function disrupts the Golgi architecture due to lack of SCYL1-COPI recruitment to the TGN, leading to Golgi enlargement and loss of protein glycosylation[36]. Human diseases associated with loss of *GORAB* functions are distinguishable from *SCYL1*-associated diseases, and include *Geroderma Osteodysplasticum*[61] and *Cutis Laxa*[62], both of which are linked to defects in extracellular matrix composition and protein glycosylation. Thus, although SCYL1 and GORAB are strong interactors, their functions can be discriminated, and mTORC1-dependent phospho-SCYL1 activities can be considered functionally distinct from the GORAB-dependent SCYL1 functions, possibly affecting a distinct SCYL1 pool.

Identification of *SCYL1* mutations in individuals suffering from CALFAN syndrome underscores the physiological relevance of this protein[29–33]. Given its function in the secretory and endolysosomal systems, it is not surprising that secretory active organs like liver and brain appear primarily affected. EVs are released in the extracellular medium and can distribute in tissue by diffusion; they can be recovered in different body fluids, including serum, urine, cerebrospinal fluid and bile, among others. EV secretion is particularly important for liver physiology and pathology[63], and may affect age-related neurodegenerative diseases[64]. Loss of *SCYL1* function will likely affect EV release in these organs, and it would be interesting to address this question by analyzing body fluids of CALFAN syndrome patients. So far, the 11 detected human mutations led to premature stop codons, loss-of-functions, or reduced protein levels in patients[30,65]. No mutation occurring at Ser754 is currently known. According to our observations, a point mutation leading to a non-phosphorylatable amino acid variant should mimic loss-of-function phenotypes.

In breast cancer, differences in SCYL1 abundance in healthy and cancer cells are being discussed that might or might not affect the stability of the REST tumor suppressor[34,35]. Whether the scaffolding functions of SCYL1 and its role in vesicle trafficking and secretion might affect cancer progression is not known. However, EV secretion by breast cancer cells was shown to promote cell-to-cell communication between tumor cells and cells of the micro-environment. In particular, EV secretion promotes tumor cell proliferation, metastasis, and evasion from the immune system and has been identified as an important mechanism contributing to the pathology of breast cancer (reviewed in ref. 66). Our findings indicate that mTORC1 and SCYL1 might regulate EV secretion in a previously unanticipated role in cancer progression.

Taken together, in the current study we characterize SCYL1 as an mTORC1 substrate and outline how mTORC1 regulates Golgi architecture, subcellular distribution of the endolysosomal system, and EV release via SCYL1 phosphorylation. Inactivation of mTORC1 does lead to a redistribution of endosomes and lysosomes and does not only induce lysosomal degradation via increased autophagy, but also increases EV release via dephosphorylation of SCYL1. Thus, organelle repositioning and regulation of specific membrane transport routes is part of the cell adaptation to nutrient availability mastered by mTORC1. Whether specific cargo molecules are differentially sorted into EVs and autophagosomes, and if this is of physiological relevance, are currently unanswered questions.

## Methods

### Cell culture and treatments

MCF-7, HEK293T, A549, and HeLa cells (ATCC® HTB-22, CRL-11268, CCL-185, and CCL-2, respectively, authenticated by genotyping (Microsynth) and frequently tested negative for mycoplasma), and RPE-1 cells were cultured in DMEM (PAN Biotech, P04-04510) supplemented with 10% fetal bovine serum (FBS, BioWest, S181B-500), 100 units/ml penicillin and 0.1 mg/ml streptomycin (PAN Biotech, P06-07100). For SILAC labeling, MCF-7 cells were cultured in parallel with differentially labeled SILAC Arg-free and Lys-free DMEM (PAN Biotech, P04-02505), supplemented with 10% dialyzed FBS (BioWest, S181D-500), 2 mM L-alanyl-L-glutamine dipeptide (GlutaMAX, ThermoFisher, 35050038), as well as L-arginine (Arg0, Fluka, 11010) and L-lysine (Lys0, Sigma-Aldrich, L8862), L-lysine:2HCl-4,4,5,5-D4 (Lys4, CIL, DLM-2640-PK), and L-arginine:HCl-13C6 (Arg6, CIL, CLM-2265-H-PK), or L-lysine:2HCl-13C6-15N2 (Lys8, CIL, CNLM-291-H-PK) and L-arginine:HCl-13C6-15N4 (Arg10, CIL, CNLM-539-H-PK), to generate "light-," "medium-," and "heavy-" labeled cells, respectively. All isotopes were added at a final concentration of 200 μM for L-arginine and 400 μM for L-lysine. All SILAC media were supplemented with an excess of 230 μM L-proline (ProO, Fluka, 81709) to avoid arginine to proline conversion. SILAC labeling was performed for at least 2 weeks and a minimum of five passages to ensure maximal isotope incorporation.

For treatments, equal amounts of cells were seeded to reach 80% confluency and treated the next day for the times described, with 100 nM rapamycin, 2 nM concanamycin A, 1 to 2 μg/ml doxycycline, 50 μg/ml cycloheximide, or 250 μM Torin1, as indicated (Sigma-Aldrich, R0395, C9705, D9891, and C6255, and TOCRIS 4247, respectively). Starvation was performed after washing cells two times in Dulbecco's phosphate buffer saline (DPBS) 1x (PAN Biotech, P04-36500) by incubation in Hank's balanced salt solution (HBSS) 1x (ThermoFisher, 14025-100) for the indicated times.

### Plasmid constructs

The following constructs were obtained from Addgene: pX330-U6-Chimeric_BB-CBh-hSpCas9 (Plasmid #42230), psPAX2 (Plasmid #12260), pMD2.G (Plasmid #12259), pCW57.1 (Plasmid #41393), pcDNA3-Flag-MTOR (Plasmid #26603). pER127, pEGFP-C1-Puro, and pBABE-myc-BirA-LC3 were generous gifts from Dr. Richard Iggo (Institut Bergonié, France), Dr. Heike Nägele (University of Freiburg, Germany), and Dr. Jay Debnath (University of California at San

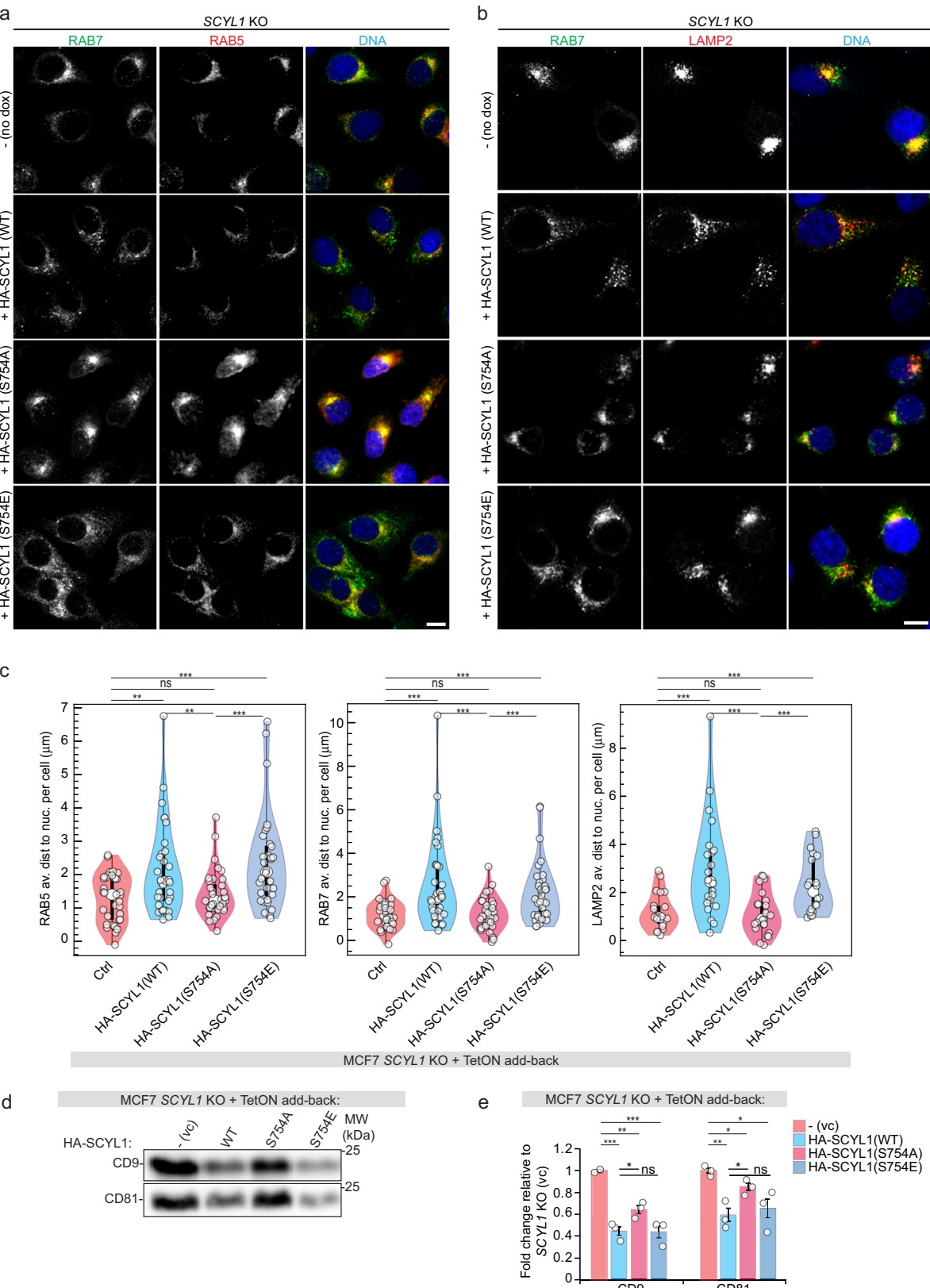

Francisco, USA), respectively. *SCYL1 KO* CRISPR/Cas9 constructs pSKP-001 and pSKP-003, and *GORAB* KO CRISPR/Cas9 constructs pSKP-127 and pSKP-128, were obtained by cloning double-stranded oligonu-cleotides oSKP-001/002 (*SCYL1* exon 1), oSKP-005/006 (*SCYL1* exon 3), oSKP-170/171 (GORAB exon2) and oSKP-172/173 (GORAB exon2'), respectively, into pX330 BbsI site (BbsI/BpiI, ThermoScientific,

ER1012). Inducible lentiviral vectors for exogenous expression of HA-SCYL1(WT) (pSKP-58), HA-SCYL1(S754A) (pSKP-136) or HA-SCYL1(S754E) (pSKP-139) recombinant proteins were obtained as fol-lows. Initially, HA-SCYL1(WT) was amplified by RT-PCR from HEK293T cDNA (Qiagen QuantiTect RT-PCR kit, 205311) using oSKP-17/21 pri-mers, and cloned into pLVXpuro (Clontech, 632164) EcoRI/XbaI sites

**Fig. 6 | SCYL1 Ser754 phosphorylation regulates endosome distribution and EV secretion. a, b** Expression of SCYL1(S754E) complements *SCYL1* KO effect on endosome distribution. Fluorescent micrograph of *SCYL1 KO* MCF-7 cells expressing HA-SCYL1(WT), HA-SCYL1(S754A) or HA-SCYL1(S754E) variants upon 24 h (for RAB5 and RAB7) and 36 h doxycycline induction (for LAMP2), respectively. Representative pictures of $n = 3$ replicates are shown. Scale bar = 10 µm.
**c** Quantification of **a, b**. The distance from nuclear envelope to RAB5, RAB7 and LAMP2 foci was computed and averaged per individual cell. All results were assembled in a violin plot (white dots within violins denote medians; top and bottom of violins represent the 25th and 75th percentiles; thick vertical lines within violins show the IQR, whereas thin vertical lines represent the rest of the distribution extending the 1.5xIQR). Single-cell values are indicated with white small dots. An unpaired two-tailed Student's *t*-test was used for comparison; RAB5: WT/VC: $p = 0.0021$ (\*\*); S754E/VC: $p = 0.00022$ (\*\*\*); WT/S754A: $p = 0.0058$ (\*\*); S754E/S754A: $p = 0.00059$ (\*\*\*). RAB7: WT/VC: $p = 0.00077$ (\*\*\*); S754E/VC: $p = 0.00057$ (\*\*\*); WT/S754A: $p = 0.00085$ (\*\*\*); S754E/S754A: $p = 0.00082$ (\*\*\*). LAMP2: WT/VC:

$p = 0.00074$ (\*\*\*); S754E/VC: $p = 0.00099$ (\*\*\*); WT/S754A: $p = 0.00039$ (\*\*\*); S754E/S754A: $p = 0.0007$ (\*\*\*). ≥20 cells were measured amongst technical triplicates of two biological replicates ($n = 6$). Error bars = SEM. Source data are provided as Source Data file. **d** HA-SCYL1 WT and S754E, but not S754A, rescue *SCYL1* KO-induced EV secretion. Western blot of ultracentrifuged CM of *SCYL1* KO MCF-7 cells expressing HA-SCYL1(WT), HA-SCYL1(S754A) or HA-SCYL1(S754E) mutants after 24 h doxycycline induction. Cells containing a vector-only transgene (VC) were used as negative control. To normalize protein contents, equal amounts of cells were used. One representative experiment out of $n = 3$ biological replicates is shown. **e** Quantification of **d**, average of $n = 3$ biological replicates. Error bars = SEM. An unpaired two-tailed Student's *t*-test was used for comparison: WT/VC: CD9 $p = 0.00031$ (\*\*\*), CD81 $p = 0.0060$ (\*\*); S754A/VC: CD9 $p = 0.0012$(\*\*), CD81 $p = 0.038$(\*); S754E/VC: CD9 $p = 0.0010$ (\*\*), CD81 $p = 0.036$ (\*); WT/S754A: CD9 $p = 0.036$(\*), CD81 $p = 0.035$(\*); WT/S754E: CD9 $p = 0.91$(ns), CD81 $p = 0.66$ (ns). All source data are provided as Source Data file.

to give pSKP-10 (pLVXpuro- HA-SCYL1(WT)). Site mutagenesis was obtained by sub-cloning PCR fragments encoding for the SCYL1 mutations serine-to-alanine (S754A) or serine-to-glutamate (S754E) were amplified from pSKP-10 using oSKP-165/166 (S754A) or oSKP-165/167 (S754E), respectively, into pSKP-10 XhoI/BsrGI sites, to generate pSKP-119 (pLVX-puro-HA-SCYL1(S754A)) and pSKP-120 (pLVX-puro-HA-SCYL1(S754E)). The inducible lentiviral vectors pSKP-58, pSKP-136 and pSKP-139 were then obtained by two-step Gateway cloning technology (Life technologies/ThermoFisher). First, a Gateway BP cloning reaction (ThermoFisher, 10348102) allowed the transfer of the WT and mutant HA-SCYL1 variants, amplified by PCR using oSKP-50/51, into pDONR201 (ThermoFisher), to give pSKP-26 (pENTR-HA-SCYL1(WT)), pSKP-130 (pENTR-HA-SCYL1(S754A)), and pSKP-131 (pENTR-HA-SCYL1(S754E)). The HA-SCYL1 variants were then transferred into the lentiviral donor vector pSKP-32 (pCW57.1 MND-Blast) by Gateway LR reaction (ThermoFisher, 10134992), to generate pSKP-58, pSKP-136 and pSKP-139. pSKP-32 (pCW57.1 MND-Blast) was obtained by replacing the AgeI/XbaI hPGK-Puromycin resistance cassette of pCW57.1 by a PCR product of the MND-Blasticidin resistance cassette, amplified from pER127 using oSKP-58/59 primers, via Gibson Assembly (New England Biolabs, E2611S), following manufacturer's instructions. The pSKP-152 Flag-MTOR inducible expression lentivirus was obtained by two-step Gateway© cloning technology as described above, with the following modifications. The Flag-MTOR ORF was amplified by PCR using pcDNA3-Flag-MTOR as template and the oSKP-190/179 primers, and inserted by BP reaction into pDONR201 to give pSKP-150 (pENTR Flag-MTOR). The Flag-MTOR ORF was then transferred by LR reaction into pCW57.1, to give pSKP-152. The vector-only control pSKP-60 construct was obtained by self-ligation of BsrGI-digested pSKP-32.

BioID inducible lentiviral constructs were generated as follows. A BirA-myc PCR fragment was generated using pBABE-myc-LC3-BirA as template and oNDS-29/42 as primers, and mixed with pSKP32 vector restriction-digested between NheI and PstI sites, to be assembled by Gibson assembly as described above and generate pDS20 (BirA-myc control). SCYL1 fusions were PCR amplified from pSKP-26, pSKP-130, and pSKP-131 using primers oNDS39/40, and mixed with BirA-myc PCR and pSKP-32 NheI/PstI to be assembled by Gibson assembly and generate pDS21 (SCYL1(WT)-BirA-myc), pDS22 (SCYL1(S754A)-BirA-myc) and pDS23(SCYL1(S754E)-BirA-myc), respectively. All PCR reactions were performed using high-fidelity enzymes Phusion or Q5 (New England Biolabs, M0530S and M0491S, respectively), according to manufacturer's instructions. All plasmids were verified by DNA sequencing (Eurofins). Sequencing reactions are provided in the Source data file under the "DNA Sequences" sheet tab.

Sequences of the oligonucleotides described above were (5' to 3'): oSKP-001, caccgacgatgtggttctttgccc; oSKP-002, aaacgggcaaagaaccac atcgtc; oSKP-005, caccgacagaggctgtgacccccgt; oSKP-006, aaacacgggg tcacagcctctgtc; oSKP-50, ggggacaagtttgtacaaaaaagcaggctatgtacccatc

gatgttccagattacgccatgtggttctttgcccgggac; oSKP-51, ggggaccactttgtaca agaaagctgggtttagtccagcttccgggctccc; oSKP-165, atgatgtacaagtccggact cagaagcagtgctcaagcttcgaattctatg; oSKP-166, ttcactcgagtccttcccaattgtc ttcgccccacgcgtctggcctcggctgggtg; oSKP-167, ttcactcgagtccttcccaattgt cttcgccccattcgtctggcctcggctgggtg; oSKP-170, caccgcagtcgcggctgcga gatt; oSKP-171, aaacaatctcgcagccgcgactgc; oSKP-172, caccgttgtcgacttt tcttcgcg; oSKP-173, aaaccgcgaagaaaagtcgacaac; oSKP-179, GGGGAC-CACTTTGTACAAGAAAGCTGGGTttaccagaaagggcaccagcc; oSKP-190, GGGGACAAGTTTGTACAAAAAAGCAGGCTgccaccatggactacaaggatgac gatgacaaggtggtggtggtggtgcgtcgactatgcttggaaccggacctgc; oNDS-29, GGTGGCGGTGGATCCGGTGGCGGTGGATCCatggacaaggacaacaccgtg cc; oNDS42, TCCAGTCACTATGGTCGACCTGCAGttattaCAGATCTTCT TCAGAAATAAGTTTTTGTTCcttctctgcgcttctcagggagatttc; oNDS-39, GTCAGATCGCCTGGAGAATTGGCTAGCgccaccATGTGGTTCTTTGCCC GGG; oNDS-40, GGATCCACCGCCACCGGATCCACCGCCACCGTCCAG CTTCCGGGCTCCC.

**CRISPR/Cas9 generation of KO clones**

*SCYL1 KO* and *GORABKO* MCF-7 cells were generated by co-transfecting pX330-derived constructs (pSKP-001 or pSKP-003 for *SCYL1 KO*, pSKP-127 or pSKP-128 for *GORAB KO*) with pEGFP-C1-Puro for selection, using lipofectamine LTX reagent (ThermoFisher, 15338030), according to manufacturer's instructions. Twenty-four hours post transfection, co-transfected cells were selected in culture medium supplemented with 3 µg/ml puromycin (Invivogen, ant-pr-1), until selection was complete. Efficiently transfected cells were isolated to generate clonal lineages by single-cell cloning in 96-well plates; colonies were all evaluated for KO efficiency by western blotting against the targeted protein. Only clones with 100% KO efficiency were conserved.

**Lentiviruses: transfection into packaging cells and target cell infection**

Lentiviral particles were produced in BSL-2 laboratory and safety cabinet, according to established protocols[67], with modifications. Briefly, lentiviral constructs containing the genes of interest (pSKP-58, pSKP-136, pSKP-139, pSKP-152, and vector control pSKP-60) were co-transfected into HEK293T cells seeded the night before with psPAX2 and pMD2.G packaging constructs, using JetPrime transfection reagent (Polyplus, 114-75). Transfection medium was changed 12 h post transfection, and lentiviral supernatants were harvested 24 h later, filter-sterilized through 0.2 µm syringe filter, supplemented with 8 µg/ml polybrene (Sigma-Aldrich, H9268), and stored in aliquots at −80 °C. Target MCF-7 *SCYL1 KO* (clone 3A1) cells and HeLa cells were seeded at 30–50% confluence 24 h before infection. To infect, viral dilutions (1:2 to 1:1000) were used in DMEM supplemented with 8 µg/ml polybrene. Infected cells were selected 24 h post infection in 4 µg/ml Blasticidin (Invivogen, ant-bl-1) or 2–3 µg/ml Puromycin, until selection was

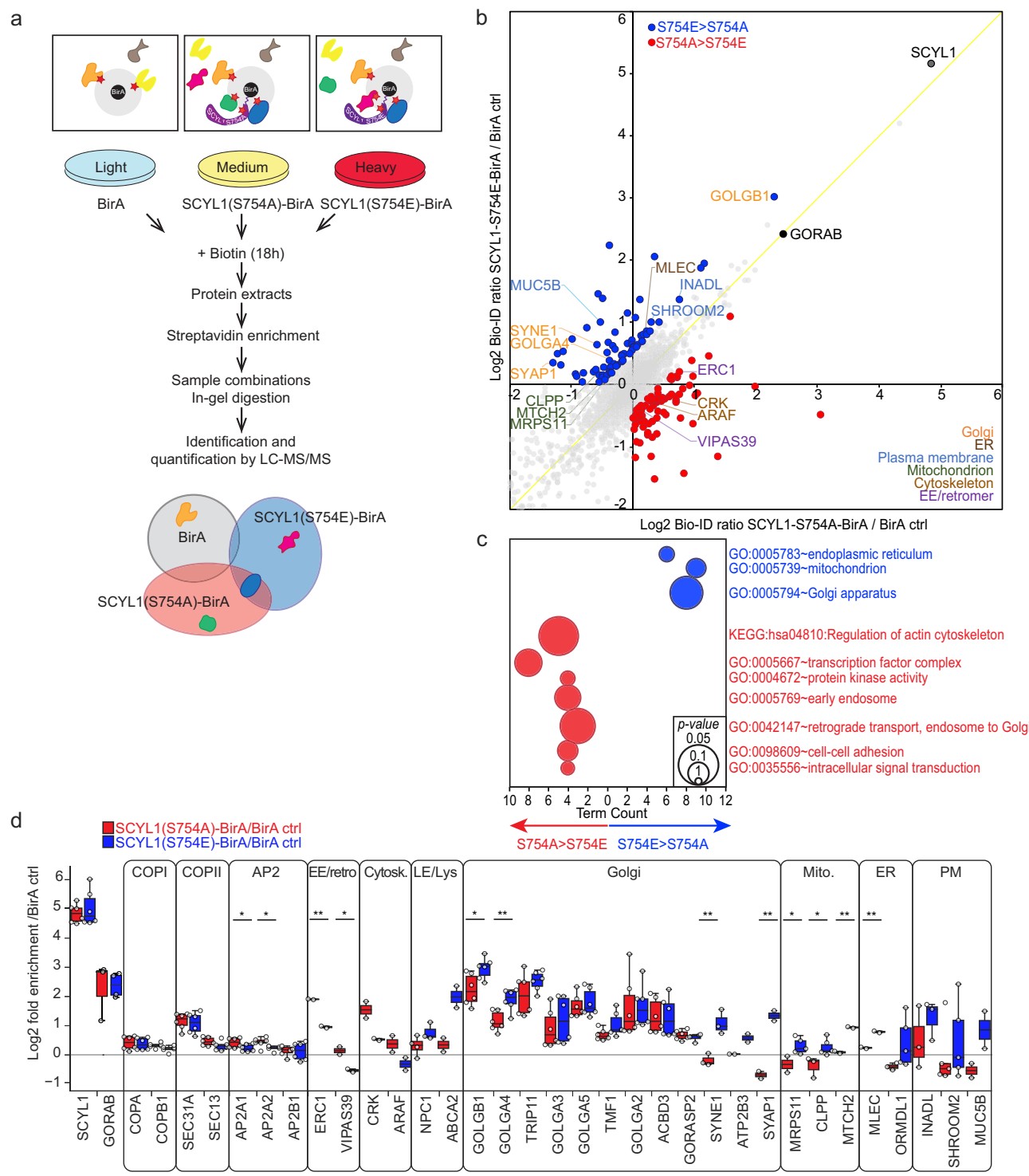

complete. Selected cells were tested by western blotting against HA-SCYL1 or FLAG-MTOR to determine the best working dilution of the lentiviral solutions.

## Protein extraction and western blotting

Upon treatment completion, cells were washed on ice twice with ice-cold 1x Dubelcco's phosphate-buffered saline (PBS, PAN biotech X0515-500), harvested by scraping, and pelleted at 300 x $g$ for 2 min in a tabletop centrifuge. Cell pellets were lysed on ice in modified RIPA buffer (50 mM Tris-HCl pH 7.5, 150 mM NaCl, 1% NP-40, 0.25% sodium deoxycholate, 1 mM EDTA) supplemented with fresh Complete protease inhibitor tablets (Sigma-Aldrich, 5056489001), and with

PhoSTOP phosphatase inhibitor if phosphorylated proteins were analyzed (Sigma-Aldrich, 4906837001). Lysates were cleared by centrifugation, quantified by BCA assay (Thermofisher, 23225), and denatured in 1x sample buffer (6 mM Tris-HCl pH 6.8, 2% SDS, 2% glycerol, bromophenol blue) at 75 °C for 10 min. For western blotting, equal amounts (30 µg) of proteins were loaded per well and separated by SDS-PAGE, and transferred on 0.45 µm nitrocellulose, polyvinylidene fluoride (PVDF) or Immobilon-E membranes (Sigma-Aldrich, GE10600016, GE10600021 and IEVH85R, respectively), by semi-dry (Trans-blot turbo transfer kit, Bio Rad, 1704271) or wet (TG buffer, VWR, 0307) transfer. Membranes were blocked in 5% milk-TBST buffer (Tris Buffered Saline (TBS) containing 0.1% tween-20) for

**Fig. 7 | SCYL1 Ser754 phosphorylation defines its association to distinct compartments. a** BioID workflow. SILAC-labeled *SCYL1* KO MCF-7 cells re-expressing Tet-inducible SCYL1(S754A)-BirA or SCYL1(S754E)-BirA were used for streptavidin-based affinity enrichment compared to cells expressing BirA only as negative control. **b** SCYL1 S754A and S754E associate with different compartments. Scatter plot displaying normalized, averaged and Log$_2$-transformed SILAC ratios from *n* = 6 biological replicates each. SCYL1 and GORAB are highlighted with black dots. Gray dots: all data points; red dots: enriched proteins (Log$_2$ difference ≥0.5) in SCYL1(S754A)-BirA; blue dots: enriched proteins (Log$_2$ difference ≥0.5) in SCYL1(S754E)-BirA. IDs of Proteins belonging to cellular compartments significantly enriched in GO/KEGG term analysis in **c**, **d** are highlighted with compartment-specific colors. Yellow diagonal indicates full correspondence between both variants. Source data for this graph are available in Supplementary Data 4. **c** GO/KEGG term enrichment analysis of proteins enriched in SCYL1 S754A and S754E BioID. Bubble plot of significant GO/KEGG terms associated with differentially enriched proteins shown in **b**. Red circles: proteins enriched in SCYL1(S754A)-BirA; Blue circles: proteins enriched in SCYL1(S754E)-BirA. A full list of all enriched GO terms is available in Supplementary Data 4. GO term enrichment *p*-values were calculated via DAVID functional annotation tool EASE Score[77].

**d** SCYL1(S754A)-BirA is more proximal to cytoskeleton and early endosome, whereas SCYL1(S754E)-BirA is more proximal to late endosome/lysosome, mitochondrial and Golgi/ER proteins. Box plot showing average SILAC ratios of SCYL1(S754A)-BirA/BirA (red) and SCYL1(S754E)-BirA/BirA (blue). Individual replicates are shown as white circles. Interactors are grouped by category: EE/retro, early endosome and retrograde transport to Golgi; Cytosk, cytoskeleton; LE/Lys, Late endosome and lysosome; Mito., mitochondrion; PM, plasma membrane. All center lines within the box plots indicate median values; box limits indicate the upper and lower quartiles of the plotted values, upper and lower whiskers indicate the largest and smallest values within the 1.5x IQR. *N* = 6 biological replicates were performed, and data were presented if at least two valid Log2-transformed ratios were available (white circles). Significant difference between SCYL1(S754A)-BirA/BirA and SCYL1(S754E)-BirA/BirA are displayed: unpaired two-tailed Student's Student's *t*-test; significant *p*-values: AP2A1: *p* = 0.054(*); AP2A2, *p* = 0.028(*); ERC1, *p* = 0.0013(**); VIPAS39, *p* = 0.041(*); GOLGB1, *p* = 0.019(*); GOLGA4, *p* = 0.0049(**); SYNE1, *p* = 0.0093(**); SYAP1, *p* = 0.011(**); MRPS11, *p* = 0.053(*); CLPP, *p* = 0.018(*); MTCH2, *p* = 0.0026(**); MLEC, *p* = 0.0049(**). Source data are provided as Source Data file and in Supplementary Data 4.

1 h, incubated with the indicated primary antibodies in the same solution overnight at 4 °C, washed three times in TBST, incubated with HRP-conjugated secondary antibodies for 1 h at room temperature, washed again three times in TBST, before performing enhanced chemiluminescence (ECL) reactions using WesternBright ECL (Advansta, K-12045-D50) or SuperSignal West femto (ThermoFisher, PIER34096) reagents. Images were recorded on Li-Cor Odyssee apparatus (Li-Cor, Li-Cor Image Studio v.2.0.38), and quantified using the Fiji software. Uncropped scans of all western blots (Main and Supplementary Figures) are provided in the Source Data file.

## Cellular subfractionation

For cellular subfractionation and enrichment in membranous compartments, equal numbers of cells were seeded 24 h prior to the indicated treatments, and harvested as described above. Cell pellets were resuspended in homogenization buffer (20 mM Hepes-NaOH pH = 7.4, 0.25 M sucrose, 1 mM EDTA) supplemented with Complete protease inhibitors (Roche), and lysed in a Dounce homogenizer with tight pistil until all cells were broken. After a whole-cell lysate (WCL) was secured, cell debris and nuclei were eliminated by two successive centrifugations at 300 x *g* for 10 min at 4 °C, and supernatants were depleted from mitochondrial membranes by centrifugation at 3000 x *g* for 10 min at 4 °C. To enrich for endosomal and Golgi membranes, supernatants were then centrifuged at 17,000 x g for 15 min at 4 °C. All WCL and 17 K samples were lysed for protein extraction in 1% Triton lysis buffer (50 mM Tris-HCl pH 7.5, 150 mM NaCl, 1% Triton X-100, 1 mM EDTA) supplemented with Complete protease inhibitors (Roche). Soluble proteins were isolated and quantified as before. Thirty micrograms of protein per sample were denatured in 1x sample buffer and boiled before loading in SDS/PAGE wells

## BioID: proximity protein labeling, streptavidin enrichment

MCF-7 *SCYL1* KO cells expressing inducible SCYL1-BirA WT, S754A and S754E variants or BirA only control were SILAC-labeled for seven generations, then treated with Biotin (50 μM, Sigma-Aldricht) for 18 h before harvesting. Cell pellets were collected and lysed in ice-cold RIPA buffer (50 mM Tris-HCl (pH 7.5), 150 mM NaCl, 1% TritonX-100, 0.1% SDS, 0.5% sodium deoxycholate, supplemented with benzonase 1/5000 and 1 mM PMSF), and lysates processed as before. Streptavidin enrichment was performed with high-capacity streptavidin-agarose at room temperature for 2 h. After affinity purification, beads were washed five times 1 ml of RIPA buffer, three times with TAP lysis buffer (50 mM HEPES-KOH pH 8.0, 100 mM KCl, 10% glycerol, 2 mM EDTA, 0.1% NP-40) and two times with 50 mM ammonium carbonate. Beads of respective SILAC experimental condition were mixed and

biotinylated proteins were eluted with 25 mM biotin in 150 μl of 1× sample buffer (200 mM Tris-HCl pH 6.8, 40 % glycerol, 8% SDS, 8 % β-mercaptoethanol, 0.04 % bromophenol blue) at 75 °C for 10 min. Eluted samples were processed for MS sample preparation as described below.

## Extracellular vesicles methods

Extracellular vesicles (Evs) were collected and characterized as described below. Our EV isolation and concentration procedure corresponds to category "intermediate recovery, intermediate specificity" according to 2018 MISEV guidelines[48].

**Isolation of extracellular vesicles (EVs).** To collect EVs, an equal amount (1.2 × 10$^7$ cells per 15 cm dish) of the indicated MCF-7 cells were seeded, and, once adherent to the dish surface, were incubated for 24 h in conditioned medium, consisting of standard DMEM supplemented with 10% FBS ultrafiltered (UF-FBS) with 100 kD molecular weight cutoff (MWCO) filtration devices (Vivaspin20, Sartorius, VS2042) and normal concentrations of penicillin/streptomycin. One 15 cm dish was used for ultrafiltration protocol, and two for ultracentrifugation protocol (see below). Cell supernatants were collected in 50 ml conical tubes (Falcon, 352098) and processed through a sequential centrifugation protocol to eliminate cells, debris and large vesicles: 300 x *g* for 10 min at 4 °C and 2000 x *g* for 10 min at 4 °C (Eppendorf 5810R centrifuge, swing bucket rotor A-4-81), then transferred to high-speed centrifugal tubes (ThermoScientific, Nalgene™ High-Speed Polycarbonate Round Bottom Centrifuge Tubes, 3117-0380) and centrifuged 10,000 x *g* for 30 min at 4 °C (Sorvall RC5B centrifuge, fixed-angle rotor SS34). Supernatants were then filtered through a 0.45 μm membrane (Filtropur S 0.45 μm, Sarstedt 83.1826), and extracellular vesicles concentrated by either ultrafiltration (for LC-MS/MS analyses) or by ultracentrifugation (for western blotting analyses). For ultrafiltration, supernatants (20 ml) were concentrated by centrifugation at 3000 x *g* and 4 °C (Eppendorf 5810 R centrifuge, swing bucket rotor A-4-81, ~20 min for 20 ml) on Vivaspin20 polyethersulfone (PES) filters with 100 kD MWCO (Vivaspin20, Sartorius, VS2042), washed once with 20 ml 1x PBS, and concentrated until their volume was reduced to 100 μl. Concentrated vesicles were directly lysed and denatured in 1x sample buffer containing 10 mM DTT for 10 min at 75 °C. For ultracentrifugation, extracellular vesicles were concentrated by two successive ultracentrifugations. For the first ultracentrifugation step, cellular supernatants (40 ml) previously cleared of cellular debris were transferred to 39 ml ultracentrifugal tubes (ThermoScientific, 010-0514) and centrifuged at 100,000 x *g* for 90 min at 4 °C (WX Ultra 80 centrifuge, fixed-angle rotor FL50 (096-

087051), ThermoScientific). EV-containing pellets were immediately washed with 2 ml ice-cold, sterile 1xDPBS (PAN Biotech, P04-36500), transferred to 3 ml ultracentrifuge tubes (Himac, S404332A), and re-pelleted by a second ultracentrifugation step for $100,000 \times g$ for 90 min at 4 °C (Discovery M150SE ultracentrifuge and fixed-angle rotor S100-A6, ThermoScientific). For western blotting analysis, EV pellets were directly lysed and denatured in 2x sample buffer supplemented with 10 mM DTT for 10 min at 75 °C. For nanoparticle tracking and scanning electron microscopy procedures, Evs were kept on ice in 1xDPBS and analyzed within hours post-isolation.

**EV characterization by proteomic profiling.** For EV proteomic profiling, the indicated cell lines were metabolically SILAC-labeled as described in Cell culture and reagents section. Conditioned medium (CM) consisted of light, medium and heavy SILAC DMEM supplemented as described, with the unique modification being that dialyzed FBS (dFBS) was replaced by ultrafiltered dialyzed FBS (UF-dFBS), generated by 100 kD molecular weight cutoff (MWCO) filtration devices (Vivaspin20, Sartorius, VS2042). SILAC-labeled Evs were isolated and concentrated using the ultrafiltration procedure described above, and differentially labeled supernatants were mixed (1:1:1, light:medium:heavy) before the initial centrifugation steps. EV protein lysates were processed for proteomic profiling as described in the MS-based proteomics section, and normalized to total protein content. Increase in EV production by *SCYL1* KO cells was characterized by the increased presence of characteristic EV proteins, as established by the 2018 MISEV guidelines for" protein content-based EV characterization"[48]. A summary of all EV-specific and associated proteins increased in SCYL1 KO cell supernatants is available in Supplementary Data 2[48].

**EV characterization by western blotting analysis.** Subsequently, the increase in EV presence in the indicated cellular supernatants was monitored by western blotting for two major EV components in our system, namely CD9 and CD81. The western blotting procedure is the same as described in our Protein extraction and western blotting section.

**EV characterization by scanning electron microscopy (SEM) and nanoparticule tracking analysis (NTA).** EVs were isolated from MCF-7 WT and *SCYL1* KO cells and concentrated by ultracentrifugation as described above. For SEM analysis, an aliquot of each EV fraction in PBS was fixed with same volume of 37% PFA for 30 min at 4 °C. Two microliters of stabilized EV fraction was absorbed on a carbon tab (ProSciTech, IA023) or a silicon wafer (Agar Scientific, AGG3390) for 5 min. Gold coating was added, with a thickness of 2 nm. Images were then taken on a Tescan Mira3 LM FE scanning electron microscope operating in the 0.5–30 kV range. For NTA, concentrated Evs were diluted in PBS and their size and concentration were determined using a NanoSight NS500 equipment (Malvern Panalyticals, Z-NTA Software version 2.3).

**Antibodies**
The rabbit polyclonal antibody specific for SCYL1 phosphorylated on serine 754 (phospho-S754-SCYL1) was raised and purified from synthetic peptide CQPRPDS(PO3H2)WGEDNW (Eurogentec). The antibody fraction recognizing the non-phosphorylated CQPRPDSWGEDNW peptide was used as anti-SCYL1 (total) in our assays.

For western blotting, the following primary antibodies were used at a 1:1000 dilution, unless otherwise stated: mouse anti-SCYL1 (Sigma-Aldrich, SAB1402612), mouse HRP-conjugated anti-GAPDH (Santa Cruz Biotechnologies, sc-25778HRP), mouse anti-Phospho-p70 S6 kinase (Thr389) (Cell Signaling, 9206s), rabbit anti- p70 S6 kinase (Cell Signaling, 9202s), rabbit anti-p62/SQSTM1 (Cell Signaling, 5114s), mouse anti-LC3B (Nanotools, 0231S0502), rabbit anti-GABARAPL1 (Cell

Signaling, 26632), mouse anti-CD9 (Santa Cruz Biotechnologies, sc-13118), mouse anti-CD81 (Santa Cruz Biotechnologies, sc-7637), rabbit anti-calreticulin (Cell Signaling, 2891 S), rabbit anti-GORAB (Atlas antibodies, HPA027250), mouse anti-COPB (Santa Cruz Biotechnologies, sc-393615), mouse anti-COPA (Santa Cruz Biotechnologies, sc-398099), rabbit anti-COPG1 (Atlas antibodies, HPA037866), mouse anti-CHMP4A/B (Santa Cruz Biotechnologies, sc-514869), mouse anti-CHMP5 (Santa Cruz Biotechnologies, sc-374338), mouse anti-TSG101 (Santa Cruz Biotechnologies, sc-136111), mouse anti-EEA1 (Santa Cruz Biotechnologies, sc-137130), mouse anti-LAMP2 (Santa Cruz Biotechnologies, sc-18822), mouse anti-RAB7 (abcam, ab50533), mouse anti-LAMP2 (Santa Cruz Biotechnologies, sc-18822), mouse HRP-conjugated anti-alpha-actin (Santa Cruz Biotechnologies, sc-47778HRP). Western blotting secondary antibodies: HRP-conjugated goat anti-Rabbit IgG (Jackson ImmunoResearch, 111-035-045) 1:10,000 dilution, HRP-conjugated goat anti-mouse IgG (Jackson ImmunoResearch, 115-035-062) 1:5000 dilution, anti-mouse IgG "VeriBlot" for IP secondary antibody, HRP-conjugated (Abcam, ab131368) 1:10,000 dilution, light-chain specific, HRP-conjugated AffiniPure goat anti-mouse IgG (Jackson ImmunoResearch, 115-035-174) 1:5000 dilution, light-chain specific, HRP-IgG fraction monoclonal mouse anti-rabbit (Jackson ImmunoResearch, 211-032-171),1:10,000 dilution.

Primary antibodies were used in immunostainings at 1:100 dilution, unless otherwise stated: rabbit anti-SCYL1 (Atlas antibodies, HPA015015), mouse anti-GOLGIN-97/GOLGA1 (ThermoScientific, A21270), mouse anti-LAMP2 (Santa Cruz Biotechnologies, sc-18822), rabbit anti-mTOR (Cell Signaling, 2983S), mouse anti-SNX1 (BD Biosciences, 611482), mouse anti-COPA (Santa Cruz Biotechnologies, sc-398099), rabbit anti-COPG1 (Atlas antibodies, HPA037866), rabbit anti-TGN46 (Atlas antibodies, HPA012609; Biorad, AHP500GT), mouse anti-GM130 (BD biosciences, #610823), mouse anti-RAB5 (Cell Signaling, 46449), mouse anti-RAB7 (abcam, ab137029), rabbit anti-GORAB (Atlas antibodies, HPA027250). Secondary antibodies for immunostainings were used at 1:2'000 dilution: Alexa Fluor 568-conjugated anti-rabbit IgG (H + L) (ThermoScientific, A11011), Alexa Fluor 488-conjugated anti-rabbit IgG (H + L) (ThermoScientific, A21206), Alexa Fluor 633-conjugated anti-mouse IgG (H + L) (ThermoScientific, A21050), Alexa Fluor 488-conjugated anti-mouse IgG (H + L) (ThermoScientific, A21202), Alexa Fluor 633-conjugated anti-rabbit IgG (H + L) (ThermoScientific, A21071), DyLight 405-conjugated anti-mouse IgG1 (Jackson Immunoresearch, 115-475-205) and Cy3-conjugated anti-Mouse IgG2b (Jackson Immunoresearch, 115-165-207).

**Immunostainings and fluorescence microscopy**
For immunofluorescence, the indicated cells were seeded on collagen I (ThermoFisher, A10483-01, diluted in 0.02 M acetic acid to 50 μg/ml)-coated coverslips 24 h prior to the experiments. The next day, cells were treated as indicated, and fixed using either paraformaldehyde (PFA) or methanol (MetOH), depending on the primary antibodies used. For the PFA fixation, cells were washed 6 times with PBS-0.1% Tween20 (PBST), then fixed 15 min in 4% PFA (Sigma-Aldrich, 15812-7), washed 6 times with PBST, and permeabilized 10 min with 0.1% Triton X-100 in PBS. For the MetOH fixation, cells were fixed in ice-cold 100% MetOH for 10 min. For both protocols, fixed cells were then washed 6 times in PBST, blocked in PBST containing 5% horse serum (ThermoScientific, 16050) for 30 min, and finally washed 6 times in PBST. Cells were incubated in a wet chamber with a primary antibody solution (previously diluted 1:100 in PBST-5% horse serum), overnight at 4 °C. Cells were then washed 6 times in PBST and incubated in the secondary antibody solution (diluted 1:2000 in PBST- 5% horse serum) in the dark, for 2 h at room temperature. After incubation, cells were washed 6 times with PBST, incubated in 10 μM Hoechst 33342 solution (Sigma-Aldrich, 14533) for 1 min, washed again six times, and embedded in ProLong Gold antifade reagent (ThermoFisher, P36931). Confocal imaging was performed using a Leica TCS SP5 system, controlled by

the Leica LAS (v2.7.3) software (Leica microsystems). Images were analyzed, quantified and prepared with Fiji (v.2.3.0[68],), Imaris (v.9.7.2 and v.9.8.2, Bitplane, Oxford Instruments) and Adobe Photoshop (v.22.5.1, Adobe) softwares, as detailed below.

RPE1 cells were fixed in 4% PFA/PBS, permeabilized with 0.1% Saponin, blocked with 1% BSA and 0.1% Saponin in PBS and incubated with primary antibodies at 1/100 dilution in blocking buffer, before washing in distilled water and incubating with the indicated secondary antibodies. To determine the co-localization between SCYL1 and TGN46 or SNX1, respectively, the Pearson's co-localization tool of the Volocity software suite (Perkin Elmer, Version 6.3) was used after setting uniform signal thresholds across the different conditions. Co-localization was determined across 10 images from two independent experiments. Significance was determined with an unpaired Student's *t*-test using Excel (Microsoft, v.16.58).

Measurements of SCYL1, foci distance to the nuclear surface was performed in Imaris (v.9.8.2) using the spots and surface tools in an ROI created around the cells of interest to exclude outliers from the analysis (e.g., dead cells, or cells damaged during the experiment). Firstly, the nuclear volume was first identified using the Surface tool, around the Hoechst-DNA-positive zone of the ROI. Secondly, SCYL1 foci were identified in the whole ROI volume using the Spots tool, based on the most intense SCYL1 fluorescence regions (growing spots with minimal radius of 0.2 μm and PSF 0.4 μm vertical correction applied). The shortest distance of each SCYL1 spot to the closest nucleus within the ROI volume was measured automatically by Imaris. Measurements of SCYL1 foci distance to Golgi was performed similarly, by defining GM130 staining as Golgi volume using the Surface tool, and SCYL1 puncta as spots of different sizes, as defined above. The shortest distance of each SCYL1 spot to the closest Golgi signal within the ROI volume was measured automatically by Imaris. Quantitative co-localization analysis of SCYL1 with GM130, RAB5, RAB7 and LAMP2 was performed using the Coloc extension of Imaris (v7.2.1, BITPLANE, Oxford Instruments) for intensity based co-localization. For the analysis, the intensity threshold for each channel was settled manually using the Isoline tool and a ROI was created around the cells of interest to exclude outlayers from the analysis (i.e., dead cells). The PSF and p-value thresholding were calculated automatically by the software. The thresholded Mander's coefficient rendered by the software is indicated in the corresponding figure. The Golgi/cell volume ratio was calculated using Imaris (v9.8.2, Bitplane). Firstly, the GOLGA1/GOLGIN97 signal was used to define the Golgi volume using the Surface tool. The cell volume was identified automatically using the Imaris Cell add-in module body detection option, which defines the best total volume for each cell, based on the background signal generated in the assay. Each cell was measured individually and automatically by Imaris for its total volume, and for the volume of all Golgi signal it contained. Data generated are provided as the ratio Golgi/total cell volume, per cell. For RAB5, RAB7 and LAMP2 foci distance to nuclei, the same approach as for SCYL1 spots distance to nuclei was used, with the difference that data were generated as an average "per cell" using the Imaris (v9.8.2) Cell add-in module. To avoid any analytical bias, settings used to define spots, surfaces and cells were recorded and automatically applied to all quantified slides through each experiment.

## Affinity purification

For anti-HA immunoprecipitation of HA-SCYL1(WT), two 15 cm dishes of MCF-7 *SCYL1 KO* transduced with pSKP-58 were induced with 1 μg/ml doxycycline for 48 h, harvested, and lysed in 1% Triton buffer as above, supplemented with Complete protease inhibitors and PhoSTOP when necessary. Soluble proteins were recovered and quantified by BCA assay, and immunoprecipitations were performed using anti-HA magnetic beads (ThermoFisher, 13474229). Beads coated with HA-SCYL1 complexes were washed three times with lysis buffer, and subsequently used for mTOR kinase assays.

## MTORC1 complex purification

Five 15 cm plates of HeLa cells overexpressing FLAG-MTOR were harvested after 24 h of 2 μg/ml doxycycline induction. Cells were lysed in lysis buffer (50 mM HEPES/NaOH pH 7.5, 5 mM CHAPS, 400 mM NaCl, 1 mM EDTA, 0.5 mM DTT, EDTA-free protease inhibitor and PhosSTOP (Roche)). The protein lysate was incubated with anti-Flag M2 Affinity gel (Sigma, M2220) for 2 h at 4 °C. After five washes with lysis buffer, the FLAG-MTOR-coated resin was split into two equal batches. One batch of was incubated with 12 μM wortmannin for 30 min at RT to inhibit the kinase activity.

## Filter-aided in vitro kinase assay

The HA-SCYL1-coated beads were resuspended in 200 μl of dephosphorylation buffer (50 mM HEPES pH = 7.5, 100 mM NaCl, 2 mM DTT, 0.01% NP-40 and 1 mM $MnCl_2$ pH = 7.5). 2000 units of λ-phosphatase were added to the beads, and this mixture was incubated at 30 °C for 30 min to dephosphorylate the endogenous phosphosites[69,70]. For *in house*-purified mTORC1 complex kinase assay, the second half of purified SCYL1 was not dephosphorylated as an internal standard. Subsequently, the beads were washed three times with kinase buffer (50 mM Tris-HCl (pH 7.4), 150 mM NaCl, 2 mM DTT, 1x PhosSTOP and 6.25 mM $MgCl_2$) to remove the λ-phosphatase.

The in vitro kinase assay was performed as described in ref. [53] with the following modifications. The commercial mTOR (TP320457, ORIGENE), or the *in house*-purified MTORC1 complex on anti-FLAG M2 affinity gel, and the HA-SCYL1-coated beads were added onto a 10 kD MW-cutoff filter (Sartorius) in 100 μl of kinase buffer with 1 mM γ-[$^{18}O_4$]-ATP (Cambridge Isotope Laboratory, OLM-7858), and for the the *in house*-purified MTORC1 assay 100 μg/ml of Flag peptide (Sigma, F3290) to elute the complex. These assays were incubated for 1 h at 37 °C, then stopped by the addition of 8 M urea and 1 mM DTT. mTOR was incubated with 12 μM wortmannin (PI3K inhibitor) for 30 min at room temperature, as a negative control for the above experiment.

Protein digestion prior to mass spectrometry analysis was performed overnight according to the FASP protocol[71]. The digested peptides were eluted twice with 100 μl of 50 mM ammonium bicarbonate. Eluates were then acidified with trifluoroacetic acid (TFA) to a final concentration of 1% and dried by SpeedVac prior to phosphopeptide enrichment.

## Phosphopeptide enrichment

For both in vitro and in vivo experiments, phosphopeptides were enriched using Titanium dioxide ($TiO_2$) beads (GL Sciences, GL-5020-7510), which were incubated with 300 mg/ml of lactic acid in 80% acetonitrile, 1% TFA prior to enrichment[72]. Samples were incubated with a slurry containing 2 mg of $TiO_2$ beads for 30 min at room temperature. Further, the mix containing beads were transferred onto 200 μl pipette tips blocked with C8 discs (3 M Empore, 66882-U). Tips were sequentially washed with 10% acetonitrile-1% TFA, 80% acetonitrile- 1% TFA, and LC-MS grade water. Phosphopeptides were finally eluted with 50 μl of 5% ammonia in 20% acetonitrile and 50 μl of 5% ammonia in 80% acetonitrile. Eluates from the samples were mixed and acidified with 20 μl of 10% formic acid. Samples were concentrated by vacuum concentration and resuspended in 20 μl of 0.1% formic acid prior to LC-MS/MS analysis. The tip flow-through was stored at −80 °C for non-phosphopeptide analysis.

## MS-based proteomics

LC-MS/MS measurements were performed on a qExactive HF-X mass spectrometer coupled to an EasyLC 1200 nanoflow-HPLC (Thermo-Scientific) and operated by Xcalibur v.4.1.31.9. Peptides were fractionated on a fused silica HPLC-column tip (I.D. 75 μm, New Objective, self-packed with ReproSil-Pur 120 C18-AQ, 1.9 μm (Dr. Maisch) to a length of 20 cm) using a gradient of A (0.1% formic acid in water) and B (0.1% formic acid in 80% acetonitrile in water): samples were loaded

with 0% B with a maximum pressure of 800 Bar; peptides were separated by 5%–30% B within 85 min with a flow rate of 250 nl/min. Spray voltage was set to 2.3 kV and the ion-transfer tube temperature to 250 °C; no sheath and auxiliary gas were used. Mass spectrometer was operated in the data-dependent mode; after each MS scan (mass range $m/z = 370$–1750; 120,000) a maximum of twelve MS/MS scans were performed using a normalized collision energy of 25%, a target value of 5000 and a resolution of 30,000. MS raw files were analyzed using MaxQuant (version 1.6.2.10)[73] using a Uniprot full-length homo sapiens database (March, 2016) plus common contaminants as reference. Carbamidomethylcysteine was set as fixed modification and protein amino-terminal acetylation, serine-, threonine- and tyrosine- (heavy) phosphorylation, and oxidation of methionine were set as variable modifications. The MS/MS tolerance was set to 20 ppm and three missed cleavages were allowed using trypsin/P as enzyme specificity. Peptide, site, and protein FDR based on a forward-reverse database were set to 0.01, minimum peptide length was set to 7, the minimum score for modified peptides was 40, and minimum number of peptides for identification of proteins was set to one, which must be unique. The "match-between-run" option was used with a time window of 0.7 min. MaxQuant results were analyzed using Perseus v.1.5.5.3[74] and Instant Clue v.0.10.10.20210316[75].

MS-based proteomics data have been deposited to the ProteomeXchange Consortium via the PRIDE partner repository:[76] (a) identifier PXD024399, [https://www.ebi.ac.uk/pride/archive/projects/PXD024399]; (b) identifier PXD024550, [https://www.ebi.ac.uk/pride/archive/projects/PXD024550].

## Transmission electron microscopy

For transmission electron microscopy, an equal volume of double-strength fixatives (4% paraformaldehyde, 5% glutaraldehyde in 0.1 M sodium cacodylate buffer [pH 7.4]) was added to the indicated cells for 20 min at room temperature (RT), prior to fixing the cells with one volume of 2% paraformaldehyde and 2.5% glutaraldehyde (Merck, 1.04239.0250) in 0.1 M sodium cacodylate buffer (pH 7.4, Sigma-Aldrich, 20840-100G-F) for 2 h at RT. After three washes with 0.1 M sodium cacodylate buffer (pH 7.4), cells were post-fixed in 1% OsO4, 1.5% potassium ferrocyanide (Sigma-Aldrich, 702587-50g) at 4 °C for 60 min. Next, the cell pellets were washed five times with distilled water and left sit in the last wash for 30 min before being centrifuged and resuspended in warm 2% low melting point agarose and immediately spun down. After solidification of the agar on ice, the tip containing the cells was cut into small 1 mm3 blocks and these blocks were incubated in 70% ethanol containing 0.5% uranyl acetate pH4, OVN at 4°C. The day after, the blocks were then dehydrated by immerging them into increasing amounts of ethanol (70%, 90%, 96%, and 3 times 100%) by incubation on a rotatory wheel for at least 15 min at RT for each step. These amalgamations were followed by others in 1,2-pro-pylene oxide (Merck, 8.02936)-Epon resin (3:1) for 30 min, 1,2-pro-pylene oxide -Epon resin (1:1) for 30 min, 1,2-propylene oxide-Epon (3:1) for 60 min and Epon resin overnight. The Epon solution was prepared by mixing 12 g of glycid ether 100, 8 g of 2-dodecenylsuccinic acid anhydride, 5 g of methylnadic anhydride (all from Serva, cat. Number 21045, 20755, 29452, respectively) and 560 ml of benzyldi-methylamine (Electron Microscopy Sciences, 11400-25). The Epon resin was then replaced the following day with freshly made resin and the incubation continued for 4 h at RT. After centrifugation at 3000 rpm for 10 min, the Epon resin was polymerized by heating the sample at 63 °C for 3 days. 70 nm sections were then cut using a Leica EM UC7 ultra microtome (Leica Microsystems) and transferred on Formvar carbon-coated copper grids. Sections were incubated with a lead-citrate solution (80 mM lead nitrate, 120 mM sodium citrate, pH 12) for 2 min, washed three times with distilled water and dried for 30 min. Cell sections were viewed using an 80 kV transmission electron microscope Talos F200i (ThermoFischer Scientific).

## Reporting summary

Further information on research design is available in the Nature Research Reporting Summary linked to this article.

## Data availability

The authors declare that the data supporting the findings of this study are available within the paper and its supplementary information files, or in the source data file. Source data are provided with this paper as a single Source data file. Raw confocal and electron microscopy images used in this study are available from the corresponding author upon request. The mass spectrometry data have been deposited to the ProteomeXchange Consortium via the PRIDE partner repository[76]. PXD024399; PXD024550. Used public databases were: https://www.proteinatlas.org/; https://www.proteomicsdb.org/, https://www.uniprot.org/, https://www.phosphosite.org/homeAction.action. Source data are provided with this paper.

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

## Acknowledgements

This research was supported by the Canton and the University of Fribourg (J.D.), the Swiss National Science Foundation grants 310030_184781 (J.D.) and CRSII5_189952 (F.R., J.D.), and the Novo Nordisk Foundation grant 0066384 (F.R. and M.M.). This work was part of the SKINTEGRITY.CH collaborative research project. We thank Dieter Kressler and Michael Stumpe from MAPP for technical support.

## Author contributions

Conceptualization, S.K.P. and J.D.; methodology, S.K.P., C.V., M.M., D.S.S., Z.H., C.R., E.M.M., H.Z., M.S.C., A.P.F., G.R., F.S., F.R., J.D.; investigation, S.K.P., C.V., M.M., D.S.S., Z.H., C.R., E.M.M., H.Z., M.S.C., F.S.; writing—original draft, S.K.P. and J.D.; writing, review, & editing, S.K.P., M.M., F.S., F.R., J.D.; funding acquisition, resources, & supervision, A.P.F., G.R., F.S., F.R., J.D.

## Competing interests

The authors declare no competing interests.

## Additional information

**Supplementary information** The online version contains

supplementary material available at https://doi.org/10.1038/s41467-022-32487-7.

