## [Peer Review File · Nature Communications]

mTORC1 controls Golgi architecture and vesicle secretion by phosphorylation of SCYL1Reviewers' comments:

Reviewer #1 (Remarks to the Author):

Kaeser-Pebernard et al., characterised the mTORC1 substrate SCYL1 and claimed that the mTORC1-dependent phosphorylation of SCYL1 controls Golgi architecture and exosomes secretion. Although this is a potentially novel finding, the manuscript does not provide any mechanistic insights into how this happens – besides through phosphorylation of SCYL1 – nor it puts the findings in the context of cellular outputs (e.g. mTORC1-driven cell growth). The experiments are well controlled, and the proteomics datasets are well performed although not analysed in depth.

Major point:

1. Why have the authors performed experiments in MCF7 and other cell lines and not in neuronal or liver cells where the importance of SCYL1 seems to be established? There is no rationale for this choice.
2. I found the results section very confusing: firstly, the authors define the phosphorylation of SCYL1, then they show its localization, then they talk about the SCYL1 KO, and then they go back to the role of SCYL1 phosphorylation.
3. Figure 2: the immunofluorescence experiment shows SCYL1 localization at the Golgi, but the fractionation experiment focus on the endosome fraction. This needs to be justified and the fractionation experiment needs to be repeated and controlled.
4. The section entitled “SCYL1 dephosphorylation leads to peripheral endosomal localization” is very misleading. Showing a change in the localization of SCYL1 does not mean that it has a role in endosomes redistribution (page 9).
5. The importance of the biological link between the Golgi and the endosomes is not highlighted in a clear manner for non-experts. As a consequence, the conclusion of the results section (“Thus, depending on the phosphorylation status of SCYL1, intracellular vesicle trafficking is altered, supporting either lysosomal targeting or exosome release.”) seem a bit of an overinterpretation. Which are the clear evidence to say that SCYL1 phosphorylation regulates either lysosomal targeting or exosomes release?

Minor points:

1. The introduction would benefit from a paragraph on the secretory pathways
2. Data not shown (e.g. p6, p10, discussion) should be shown.
3. Figure 1c: why to use wortammine here instead of rapamycin? The experiment needs to be repeated
4. I do not understand the rationale behind Supplementary Figure S3b-c

Reviewer #2 (Remarks to the Author):

In the reviewed manuscript entitled “mTORC1 controls Golgi architecture and exosome secretion by phosphorylation of SCYL1 ” by Kaeser-Pebernard et al. the authors present their study on SCYL1, a kinase-like protein and Golgi-localized target of mTORC1 that regulates vesicle transport and exosome release. The authors identify Ser754 as the amino acid on SCYL1 to be phosphorylated by mTORC1 that controls SCYL1 association and morphology of the Golgi apparatus. Interestingly, the observed phenotypes mimic a loss-of-function of SCYL1 is associated with a human genetic disorder called the CALFAN syndrome.

Overall, the study is well designed, the manuscript is well written, and the data are

interesting, some rather minor issues remain and need to be addressed.

Points to address:

- It is not entirely convincing that mTORC1 is the only kinase to phosphorylate SCYL1 on Ser754 given the data presented in Figure 1b. As the authors rightly state, SCYL1 at Ser754 is rapidly dephosphorylated upon mTORC1 inhibition, but this loss of phosphorylation is not complete as seen in other, well-accepted substrates, such as pS6K1. This could have to do with kinetics of dephosphorylation by an unknown phosphatase of SCYL1. Please discuss the observation that the levels of pSCYL1 are not entirely gone even after 1h of starvation (with HBSS w/o FBS) or Rapamycin treatment.
- The authors state that a subset of SCYL1 localizes to the cell periphery upon mTOR inhibition (Figure 2a and 2e). While the endosomal association of SCYL1 especially with starvation is convincing, the quantification of immunostaining data using line plots is rather not and seems somewhat biased and depends on the experimenter's choice of a region. Using other quantification methods, such as concentric circles surrounding the nucleus and quantifying fluorescence signal within these circles would be more convincing. In addition, an assessment of Mander's coefficient would be helpful here.
- Data shown in Figure 2e indicates that in addition to mTOR, there are likely other kinases regulating SCYL1 association with endosomes, assuming that inhibition of mTOR with Rapa was complete. Can the authors please assess whether this was the case in the presented experiments? If mTOR activity was completely abolished with Rapa here, please discuss this result indicating a likely involvement of additional kinases in the process.
- Please consider supplementing analyses in Figure 2e with blots for Rab5 and Rab7.
- In figure legend 4 it is not clear what analyses have been performed. Please indicate how the exosome protein profiling data was generated (likely to be proteomic analyses on culture medium from WT and KO cells).
- While discussing the involvement of signaling endosomes and different mTORC1 pools on page 18, please consider and include the study published by Nnah et al., 2019 in Autophagy. This study shows the existence of endosomal mTORC1 on signaling endosomes in mammalian cells, and role in maintaining autophagy during prolonged starvation.

Reviewer #3 (Remarks to the Author):

The authors studied the role of mTORC1 in regulating SCYL1 via S574 phosphorylation. The authors claim that SCYL1 regulates vesicular transport and extracellular vesicles release in mTORC1 dependent manner. The study encompasses cell biology, biochemical and proteomics data that forms the strength of the article. The article however over claims in several places without strong supporting data. The part involving exosomes or extracellular vesicles is poor and lacks quantitative data. The study is performed only in one cell line. Moreover, the study as presented is not coherent and seems like two stories are joined together: hence, not an intuitive read. Perhaps the authors may need to rejig as how the article is presented. Overall, the article is weak on several fronts and needs significant additional data.

1. Claim that mTORC1 is the direct kinase that phosphorylates SCYL1 at S574 is not convincing. The direct evidence provided is limited (only Fig 1c). It is unclear as how the authors performed in vitro kinase assay? Was the SCYL1 purified from bacteria? Was these

phosphorylated in S574 already? What are the controls here? If purified from mammalian cells, was it already phosphorylated in S574?

2. Can the authors perform a mTORC1 KO to validate that SCYL1 is phosphorylated by mTORC1 at S574?

3. Exosomes data is not quantitative and hence the claims are not supported well with data. These data does not provide confidence in the article.

4. Ultrafiltration (UF) was used to isolate EVs in certain cases Fig 4. These cannot be called EVs and rather a very crude prep with contaminants from conditioned media. These data need to be replaced with ultracentrifugation as UF does not provide confidence with the data.

5. Fig 4b needs more markers like alix or tsg101 or sytenin and cd63. Was the protein amount normalised to equal cell number or is it total protein? Same for 4h.

6. Was the particle number normalised against equal live cells in 4e? At the moment it does not look like that? What is the proliferation status of SCYL1 KO cells?

7. Claim that SCYL1 reduces EV release in Fig 6d lacks QUANTITATIVE data. The Western blot need to be normalised by cell number. In addition NTA data needs to be provided again normalised by equal cell number.

8. Additional cell lines need to be used to validate the important observations.

9. The use of the TERM exosomes need to be avoided as per MISEV guidelines. The authors can perhaps use small extracellular vesicles (sEV) as the have done 100K. The isolated fraction contains both endosome- and PM-derived EVs so it is wise to name them EVs.

10. Proteomics datasets need validation. For instance, several EV proteins are enriched in KO cells – can authors perform Western blot and confirm this.

11. Claims at Page 12 “. These data show that mTORC1-dependent SCYL1 Ser754 phosphorylation is necessary and sufficient to block exosome release, regardless of SCYL1 subcellular localization and the presence or absence of GORAB.” is not supported by the data and is a over claim. No data supporting this has been provided.

12. Phosphoantibody generation needs clarity. In Supplementary Fig. 1, was SCYL1 KO cells used? If not, what is the basal endogenous S574 phosphorylation status? The bands detected in S572A/E – are these endogenous levels? The authors say cross reactivity with total SCYL1 while endogenous levels never mentioned? Because the cells are not clearly mentioned, it is confusing as whether the authors used MCF7 cells or SCYL1 KO? Also, if KO cells were used – what about the leakage in the DOX system? Controls for these are missing. As authors are not providing more information in text, several questions arise. Can the authors provide more information in the main article throughout – like cells and precise condition etc.

13. Fig 1 d and e does not correlate well. Western blot shows starvation has more effect on S574-p while 1e shows no difference between starvation and rapamycin (30 min seems almost half between Rapamycin and Starvation in Western blot)

14. Please mentions cell names throughout – figure 1 description in text and legend has no cell names.

Reviewer #1 (Remarks to the Author):

Kaesler-Pebernard et al., characterised the mTORC1 substrate SCYL1 and claimed that the mTORC1-dependent phosphorylation of SCYL1 controls Golgi architecture and exosomes secretion. Although this is a potentially novel finding, the manuscript does not provide any mechanistic insights into how this happens – besides through phosphorylation of SCYL1 – nor it puts the findings in the context of cellular outputs (e.g. mTORC1-driven cell growth). The experiments are well controlled, and the proteomics datasets are well performed although not analysed in depth.

We thank the reviewer for the critical comments which we addressed as outlined in detail below.

Major point:

1. Why have the authors performed experiments in MCF7 and other cell lines and not in neuronal or liver cells where the importance of SCYL1 seems to be established? There is no rationale for this choice.

We excuse ourselves for not stating clearly why we used MCF7 cells. As indicated by the link to CALFAN syndrome, SCYL1 seems to be important in secretory organs. We checked mRNA expression levels of 69 cell lines on <https://www.proteinatlas.org/> and MCF7 cells have the highest mRNA levels of all analyzed cell lines. In addition, we checked protein abundance levels of 77 cell lines on <https://www.proteomicsdb.org/> and cell lines derived from breast exhibit the highest SCYL1 protein levels, again MCF7 cells being in the top 15%. We added a respective statement to the results section, page 5:

“To study the functions of SCYL1 in a relevant model system, we analysed mRNA and protein levels in commonly used cell lines using public databases (<https://www.proteinatlas.org/>; <https://www.proteomicsdb.org/>). The highest levels of both mRNA and protein were found in breast and mammary gland-derived cell lines, which is in line with the observations that loss-of-function mutations of SCYL1 lead to phenotypes in secretory organs. Hence, we chose MCF7 breast carcinoma cells, which express the highest amount of SCYL1 mRNA to study a potential crosstalk between SCYL1 and mTORC1.”

2. I found the results section very confusing: firstly, the authors define the phosphorylation of SCYL1, then they show its localization, then they talk about the SCYL1 KO, and then they go back to the role of SCYL1 phosphorylation.

Also, in light of comments of reviewer 3 it seems that the order of data presentation was suboptimal. We restructured the entire results section. Firstly, we study regulation of localization of endogenous SCYL1. Next, we focus on the phenotype of SCYL1 KO cells, and finally we study mTORC1 phosphorylation of SCYL1 and analyse the function of this phosphorylation.

3. Figure 2: the immunofluorescence experiment shows SCYL1 localization at the Golgi, but the fractionation experiment focus on the endosome fraction. This needs to be justified and the fractionation experiment needs to be repeated and controlled.

As suggested by the reviewer, we repeated the fractionation experiments including additional marker proteins for Golgi and endosomes. In addition, we also performed additional immunofluorescence analyses studying SCYL1-endosome localization. We included data as **new Figure 1g-j**:

Figure 1. (g) Endosomal SCYL1 increases upon mTORC1 inhibition. Cellular fractionation of WT and SCYL1 KO MCF7 cells untreated (DMEM) or treated for 3 h with 100 nM Rapamycin (Rapa) or starved in HBSS (Starv). Concanamycin A (ConA) was added in parallel to all conditions to block lysosomal degradation. To normalize protein contents, proteins within each fraction were extracted and quantified, and equal amounts were loaded per well. WCL: Whole cell lysates; 17k: 17'000xg fractionation pellet enriched in endosomal membranes. One representative experiment out of n=3 biological replicates is shown. **(h) Quantification of (g).** n=3. p-values: two-tailed Student's t-test value of the indicated data point, compared with the reference value "DMEM - conA". WCL SCYL1 protein levels were normalized to β -Actin/ACTB levels, whereas 17K SCYL1 protein levels were normalized to EEA1 levels. *: $p \leq 0.05$; **: $p \leq 0.01$; ***: $p \leq 0.001$. Error bars indicate SEM. **(i) SCYL1 foci colocalize with early and late endosome markers RAB5 and RAB7.** Representative fluorescence microscopy images from untreated WT MCF7 cells (DMEM) or

treated for 3 h with 100 nM Rapamycin (“Rapa”). Cells were immunostained for endogenous SCYL1 (green), and either early endosome marker RAB5, late endosome marker RAB7, or lysosomal marker LAMP2 (in red). Representative pictures of n=3 replicates are shown. Scale bar = 10 μ m. **(j) Colocalization quantification of (i).** Rapamycin treatment (Rapa) leads to increased colocalization between SCYL1-RAB5 and SCYL1-RAB7. LAMP2 behaves opposite. *:p<0.05, Student’s t test. Error bars indicate SEM.

4. The section entitled “SCYL1 dephosphorylation leads to peripheral endosomal localization” is very misleading. Showing a change in the localization of SCYL1 does not mean that it has a role in endosomes redistribution (page 9).

This statement was indeed misleading and referred partially to data shown in the old Figure 7. We rewrote the entire section and included now two new paragraphs:

- **mTORC1 phosphorylation of SCYL1 Ser754 controls its localization and Golgi architecture**
- **SCYL1 Ser754 phosphorylation regulates the subcellular distribution of endosomes and lysosomes as well as EV secretion**

5. The importance of the biological link between the Golgi and the endosomes is not highlighted in a clear manner for non-experts. As a consequence, the conclusion of the results section (“Thus, depending on the phosphorylation status of SCYL1, intracellular vesicle trafficking is altered, supporting either lysosomal targeting or exosome release.” seem a bit of an overinterpretation. Which are the clear evidence to say that SCYL1 phosphorylation regulates either lysosomal targeting or exosomes release?

As suggested we rephrased this statement and down-tuned our conclusions. We do not address vesicular trafficking any longer but clearly highlight the effects on endosome localization/positioning. We added a **new Figure 2g** highlighting changes in endosome positioning in *SCYL1* KO cells. This agrees to our conclusions and data presented in **new Figure 6a-b** highlighting different localization of RAB5- and RAB7-positive endosomes and LAMP2-positive lysosomes in cells expressing different SCYL1 variants. With respect to secretion, we added protein quantification of conditioned medium of cells expressing different SCYL1 variants (**new Figure 6d**).

Figure 2. (g) Loss of SCYL1 leads to a change of early and late endosome localization.

Representative fluorescence microscopy images from WT and SCYL1 KO MCF7 cells in normal growth conditions (DMEM). Cells were immunostained for early endosome (RAB5) and late endosome (RAB7) markers. Representative pictures of n=3 replicates are shown. Scale bar = 10 μ m

Figure 6: mTORC1-dependent SCYL1 Ser754 phosphorylation regulates the subcellular endosomal distribution and EV secretion. (a-b) Expression of SCYL1S754E compensates SCYL1 KO effect on endosome localization. Representative fluorescent microscopy pictures of SCYL1 KO MCF7 cells expressing HA-SCYL1WT, HA-SCYL1S754A or HA-SCYL1S754E mutants upon 24 h (for RAB5 and RAB7) and 36 h doxycycline induction (for LAMP2), respectively. Cells were immunostained for early endosome marker RAB5 and late endosome marker RAB7 in (a), or for late endosome marker RAB7 and lysosomal marker LAMP2 in (b), as indicated. The merged pictures also show DNA stained by Hoechst reagent. Representative pictures of n=3 replicates are shown. Scale bar = 10 μ m. **(c) HA-SCYL1 WT and S754E, but not S754A, rescue SCYL1 KO-induced EV secretion.** Western blot analysis of ultracentrifuged CM of SCYL1 KO MCF7 cells expressing HA-SCYL1WT, HA-SCYL1S754A or HA-SCYL1S754E mutants after 24h doxycycline induction. Cells containing a vector only transgene (VC) were used as negative control and treated in the same conditions. To normalize protein contents, equal amounts of cells were used for EV purification. One representative experiment out of n= 3 biological replicates is shown. **(d) Quantification of (c),** average of n=3 biological replicates. Error bars= SEM. *: p-value \leq 0.05; (two-tailed Student's t-test between the indicated data points).

Minor points:

1. The introduction would benefit from a paragraph on the secretory pathways

Due to the length of the manuscript, we only added a brief introduction to the secretory pathway on page 3.

“Regulation of metabolism by mTORC1 therefore seems to be intimately linked to multiple membranous compartments, including those of the secretory pathways, the canonical one leading to the secretion of signal peptide containing proteins, which reach the plasma membrane via ER insertion and COPII-coated vesicle-mediated transport via the Golgi¹⁸.”

2. Data not shown (e.g. p6, p10, discussion) should be shown.

We do show all data.

3. Figure 1c: why to use wortammine here instead of rapamycin? The experiment needs to be repeated

As these were *in vitro* experiments with purified kinase and purified substrate, rapamycin cannot be used. Rapamycin inhibits mTORC1 through the formation of an inhibitory complex with the cytosolic receptor protein FKBP12 which is not present *in vitro* (see original publication PMID: 1996117).

However, we performed additional *in vivo* experiments and also used the selective ATP-competitive mTORC1 inhibitor Torin1 and added respective data as **new Figure 4d-e**. Data agree to our conclusions. In addition, we performed refeeding experiments studying mTORC1 reactivation clearly showing that SCYL1 gets phosphorylated in a time-dependent manner, similar to the known mTORC1 target RPS6K. Data are added as **new Figure 4f-g**.

Figure 4. (d) Time-course of SCYL1 Ser754 phosphorylation upon mTORC1 inhibition, detected by a phosphosite-specific antibody. WT MCF7 cells were left untreated or treated with 100 nM

Rapamycin, starvation (HBSS), or 250 μ M Torin1, for the indicated times. One representative time-course experiment out of 3 biological replicates is shown. Equal protein amounts were loaded. **(e) Quantification of (d).** Phospho-Ser754-SCYL1 levels were normalized to total SCYL1 levels. Average of n=3 biological replicates. Red line: rapamycin treatment /DMEM; blue line: starvation treatment /DMEM; green line: Torin1 treatment /DMEM. p-values: two-tailed Student's t-test between the indicated time points and referenced to 0 min, colored by treatment as above. *: $p \leq 0.05$, **: $p \leq 0.01$. ns: non-significant. Error bars indicate SEM. **(f) SCYL1 Ser 754 phosphorylation responds to mTORC1 reactivation.** WT MCF7 cells were starved in HBSS for 4 h, then released in complete DMEM supplemented with 50 μ g/ml cycloheximide to increase the intracellular amino acid concentration for the indicated times. One representative time-course experiment out of 3 biological replicates is shown. Equal protein amounts were loaded. **(g) Quantification of (f).** Phospho-Ser754-SCYL1 levels were normalized to total SCYL1 levels. Average of n=3 biological replicates. p-values: two-tailed Student's t-test between the indicated time points and referenced to Starved cells (before release). **: $p \leq 0.01$. ns: non-significant. Error bars indicate SEM.

4. I do not understand the rationale behind Supplementary Figure S3b-c

These data were added to confirm the published effect of SCYL1 on autophagy (PMID: 21508686). So far, respective data were generated by siRNA-mediated knockdown of SCYL1. We used for the first time Crispr/Cas9-based knock out cells. Thus, we intended to show that our cells confirm published data and behave as expected. We added a respective explanatory statement.

Reviewer #2 (Remarks to the Author):

In the reviewed manuscript entitled “mTORC1 controls Golgi architecture and exosome secretion by phosphorylation of SCYL1” by Kaeser-Pebernard et al. the authors present their study on SCYL1, a kinase-like protein and Golgi-localized target of mTORC1 that regulates vesicle transport and exosome release. The authors identify Ser754 as the amino acid on SCYL1 to be phosphorylated by mTORC1 that controls SCYL1 association and morphology of the Golgi apparatus. Interestingly, the observed phenotypes mimic a loss-of-function of SCYL1 is associated with a human genetic disorder called the CALFAN syndrome.

Overall, the study is well designed, the manuscript is well written, and the data are interesting, some rather minor issues remain and need to be addressed.

We thank the reviewer for the positive feedback and the interest in our work.

Points to address:

- It is not entirely convincing that mTORC1 is the only kinase to phosphorylate SCYL1 on Ser754 given the data presented in Figure 1b. As the authors rightly state, SCYL1 at Ser754 is rapidly dephosphorylated upon mTORC1 inhibition, but this loss of phosphorylation is not complete as seen in other, well-accepted substrates, such as pS6K1. This could have to do with kinetics of dephosphorylation by an unknown phosphatase of SCYL1. Please discuss the observation that the levels of pSCYL1 are not entirely gone even after 1h of starvation (with HBSS w/o FBS) or Rapamycin treatment.

This is a very relevant comment and we share the opinion of the reviewer. **We added respective statements to the results and discussion.** In addition, we performed refeeding experiments highlighting the time-dependent phosphorylation of SCYL1 (**new Figure 4f-g**). Also in these experiments, a residual SCYL1 phosphorylation is visible indicating that additional kinases might phosphorylate SCYL1 Ser754 (or that the site is particularly resistant to phosphatases).

Figure 4. (f) SCYL1 Ser 754 phosphorylation responds to mTORC1 reactivation. WT MCF7 cells were starved in HBSS for 4 h, then released in complete DMEM supplemented with 50 µg/ml cycloheximide to increase the intracellular amino

acid concentration for the indicated times. One representative time-course experiment out of 3 biological replicates is shown. Equal protein amounts were loaded. **(g) Quantification of (f).** Phospho-Ser754-SCYL1 levels were normalized to total SCYL1 levels. Average of n=3 biological replicates. p-values: two-tailed Student’s t-test between the indicated time points and referenced to Starved cells (before release). **: p ≤ 0.01. ns: non-significant. Error bars indicate SEM.

- The authors state that a subset of SCYL1 localizes to the cell periphery upon mTOR inhibition (Figure 2a and 2e). While the endosomal association of SCYL1 especially with starvation is convincing, the quantification of immunostaining data using line plots is rather not and seems somewhat biased and depends on the experimenter's choice of a region. Using other quantification methods, such as concentric circles surrounding the nucleus and quantifying fluorescence signal within these circles would be more convincing. In addition, an assessment of Mander's coefficient would be helpful here.

As suggested, we reanalyze our data and performed additional experiments quantifying SCYL1 subcellular localizations. We added data as **new Figure 1**.

Figure 1: (c) The COPI-coat complex is redistributed and remains associated with SCYL1 at cytoplasmic locations in mTORC1-inhibited conditions. Representative fluorescence microscopy images from WT MCF7 cells untreated (DMEM) or treated for 3 h with 100 nM Rapamycin (“Rapa”). Cells were immunostained for endogenous SCYL1 (red), COPI member COPA (green) and Cis-Golgi marker GM130/GOLGA2 (blue). Yellow lines: line drawings used to generate plot profiles in (d). Representative pictures of n=3 replicates are shown. Scale bar = 10 µm.

(d) Plot profiles of (c). Profiles represent the percentage of maximal intensity in function of the distance along selected plot profile lines for SCYL1 (red), COPA (green) and GM130/GOLGA2 (blue). **(e) SCYL1 distance to Golgi increases upon Rapamycin treatment.** SCYL1 puncta distance to GM130-positive Golgi edge as in (c) was measured and compared between DMEM and 3 h Rapamycin-treated conditions. The percentage of SCYL1 puncta in close proximity (between 0 and 1 µm from Golgi edge, grey boxes) was distinguished from the percentage of puncta located further away from the Golgi edge (> 1 µm and up to 21 µm as observed maximum, white boxes). A distance of 0 µm indicates colocalization with Golgi. **: p≤0.01, Student's t test, n=5 replicates (SCYL1 puncta from >10 cells measured per replicate). Error bars indicate SEM. **(f) Colocalization quantification of (c).** Rapamycin treatment (Rapa) leads to increased colocalization between SCYL1 and COPA. **: p≤0.01, Student's t test, n=4. Error bars indicate SEM.

Figure 1. (i) SCYL1 foci colocalize with early and late endosome markers RAB5 and RAB7. Representative fluorescence microscopy images from untreated WT MCF7 cells (DMEM) or treated for 3 h with 100 nM Rapamycin (“Rapa”). Cells were immunostained for endogenous SCYL1 (green), and either early endosome marker RAB5, late endosome marker RAB7, or lysosomal marker LAMP2 (in red). Representative pictures of n=3 replicates are shown. Scale bar = 10 μ m. **(j) Colocalization quantification of (i).** Rapamycin treatment (Rapa) leads to increased colocalization between SCYL1-RAB5 and SCYL1-RAB7. LAMP2 behaves opposite. *:p<0.05, Student’s t test. Error bars indicate SEM.

- Data shown in Figure 2e indicates that in addition to mTOR, there are likely other kinases regulating SCYL1 association with endosomes, assuming that inhibition of mTOR with Rapa was complete. Can the authors please assess whether this was the case in the presented experiments? If mTOR activity was completely abolished with Rapa here, please discuss this result indicating a likely involvement of additional kinases in the process.

We do indeed have data highlighting that mTORC1 inhibition is rather complete as judged by the phosphorylation level of its direct downstream target ribosomal S6 kinase (see data presented above). We discuss these findings as suggested.

- Please consider supplementing analyses in Figure 2e with blots for Rab5 and Rab7.

As suggested we added RAB7. As EEA1 is addressing early endosomes, we added the Golgi marker GOLGA1 (new Figure 1g-h).

Figure 1. (g) Endosomal SCYL1 increases upon mTORC1 inhibition. Cellular fractionation of WT and SCYL1 KO MCF7 cells untreated (DMEM) or treated for 3 h with 100 nM Rapamycin (Rapa) or starved in HBSS (Starv). Concanamycin A (ConA) was added in parallel to all conditions to block lysosomal degradation. To normalize protein contents, proteins within each fraction were extracted and quantified, and equal amounts were loaded per well. WCL: Whole cell lysates; 17k: 17'000xg fractionation pellet enriched in endosomal membranes. One representative experiment out of n=3 biological replicates is shown. **(h) Quantification of (g).** n=3. p-values: two-tailed Student's t-test value of the indicated data point, compared with the reference value "DMEM - conA". WCL SCYL1 protein levels were normalized to β -Actin/ACTB levels, whereas 17K SCYL1 protein levels were normalized to EEA1 levels. *: $p \leq 0.05$; **: $p \leq 0.01$; ***: $p \leq 0.001$. Error bars indicate SEM.

- In figure legend 4 it is not clear what analyses have been performed. Please indicate how the exosome protein profiling data was generated (likely to be proteomic analyses on culture medium from WT and KO cells).

Respective statements were presented in the methods section. We added all relevant details to the results section and figure legend.

- While discussing the involvement of signaling endosomes and different mTORC1 pools on page 18, please consider and include the study published by Nnah et al., 2019 in Autophagy. This study shows the existence of endosomal mTORC1 on signaling endosomes in mammalian cells, and role in maintaining autophagy during prolonged starvation.

We discussed the respective manuscript added the reference.

Reviewer #3 (Remarks to the Author):

The authors studied the role of mTORC1 in regulating SCYL1 via S574 phosphorylation. The authors claim that SCYL1 regulates vesicular transport and extracellular vesicles release in mTORC1 dependent manner. The study encompasses cell biology, biochemical and proteomics data that forms the strength of the article. The article however over claims in several places without strong supporting data. The part involving exosomes or extracellular vesicles is poor and lacks quantitative data. The study is performed only in one cell line. Moreover, the study as presented is not coherent and seems like two stories are joined together: hence, not an intuitive read. Perhaps the authors may need to rejig as how the article is presented. Overall, the article is weak on several fronts and needs significant additional data.

We thank the reviewer for reading and commenting on our paper. Also, in light of comments of reviewer 1 it seems that the order of data presentation was suboptimal. We restructured the entire results section. Firstly, we study regulation of localization of endogenous SCYL1. Next, we focus on the phenotype of SCYL1 KO cells, and finally we study mTORC1 phosphorylation of SCYL1 and analyse the function of this phosphorylation.

However, we are surprised by the comment that our analyses lack quantitative data. In the main figures as well as in the supplemental figures and tables detailed quantitative data are presented. We might not have highlighted this clearly in the submitted manuscript and changed this in the revised version (please see detailed comments below).

1. Claim that mTORC1 is the direct kinase that phosphorylates SCYL1 at S574 is not convincing. The direct evidence provided is limited (only Fig 1c). It is unclear as how the authors performed *in vitro* kinase assay? Was the SCYL1 purified from bacteria? Was these phosphorylated in S574 already? What are the controls here? If purified from mammalian cells, was it already phosphorylated in S574?

We did perform two types of *in vitro* kinase assays as stated in the original version of the manuscript. To highlight this more clearly, we rewrote this part and added experimental details that were included in the methods section of the original manuscript.

“To establish whether Ser754 is a direct mTORC1 phosphosite, we performed *in-vitro* kinase assays. Briefly, immunoprecipitated HA-SCYL1 was dephosphorylated by λ -phosphatase to remove endogenous phosphate groups. Purified HA-SCYL1 was then incubated **with either** the kinase subunit MTOR purified from HEK293T cells (see Methods for details), **or** entire mTORC1 complex (consisting of MTOR, LST8 and RPTOR) immunoprecipitated from HeLa cells overexpressing FLAG-MTOR. These experiments were carried out in both the absence and the presence of wortmannin, a well-characterized phosphatidylinositol-3-kinase inhibitor that potently blocks mTORC1^{45, 46}. To discriminate phosphatase resistant sites from sites phosphorylated *in vitro*, γ -1804-labeled ATP was used in the reaction. Phosphorylated peptides marked by heavy phosphate were quantified by LC-MS/MS.”

2. Can the authors perform a mTORC1 KO to validate that SCYL1 is phosphorylated by mTORC1 at S574?

As far as we know mTOR kinase KO are not viable. We could generate a RPTOR KO which disrupts mTORC1 leaving mTORC2 intact. However, it was already shown that mTORC2 might compensate for the loss of mTORC1, thus we are not sure how interpretable respective results would be. Rather we performed additional in vivo experiments (a) using Torin1 as pharmacological mTORC1 inhibitor, and (b) performing refeeding experiments to study the kinetics of SCYL1 phosphorylation. Data was added as **new Figure 4d-g**.

Figure 4. (d) Time-course of SCYL1 Ser754 phosphorylation upon mTORC1 inhibition, detected by a phosphosite-specific antibody. WT MCF7 cells were left untreated or treated with 100 nM Rapamycin, starvation (HBSS), or 250 μ M Torin1, for the indicated times. One representative time-course experiment out of 3 biological replicates is shown. Equal protein amounts were loaded. **(e) Quantification of (d).** Phospho-Ser754-SCYL1 levels were normalized to total SCYL1 levels. Average of $n=3$ biological replicates. Red line: rapamycin treatment /DMEM; blue line: starvation treatment /DMEM; green line: Torin1 treatment /DMEM. p-values: two-tailed Student's t-test between the indicated time points and referenced to 0 min, colored by treatment as above. *: $p \leq 0.05$, **: $p \leq 0.01$. ns: non-significant. Error bars indicate SEM. **(f) SCYL1 Ser 754 phosphorylation responds to mTORC1 reactivation.** WT MCF7 cells were starved in HBSS for 4 h, then released in complete DMEM supplemented with 50 μ g/ml cycloheximide to increase the intracellular amino acid concentration for the indicated times. One representative time-course experiment out of 3 biological replicates is shown. Equal protein amounts were loaded. **(g) Quantification of (f).** Phospho-Ser754-SCYL1 levels were normalized to total SCYL1 levels. Average of $n=3$ biological replicates. p-values: two-tailed Student's t-test between the indicated time points and referenced to Starved cells (before release). **: $p \leq 0.01$. ns: non-significant. Error bars indicate SEM.

3. Exosomes data is not quantitative and hence the claims are not supported well with data. These data does not provide confidence in the article.

We are not sure what the reviewer is referring to? We quantified ultra-filtered (100 kDa cutoff) conditioned medium of WT and KO cells using MS-based proteomics (old Figure 4a) and added the respective data as supplemental table S2 (old Table S3). All proteins are listed with their abundance ratios and the % variability, i.e. the accuracy of quantification. **All experiments directly addressing vesicles were performed by using ultracentrifuged CM.** In old Figure 4e we used a Nanoparticle Tracking Assay (NTA) to count and quantify extracellular vesicle sizes. Six technical replicates were performed per biological replicate and error of measurements are presented as SEM for each data point.

If the reviewer is referring to old Figure 4h: these blots were meant to validate our proteomics data presented in old Figures 4a and 4g. We performed additional proteomic experiments and added quantification and statistical analyses as **new Figure 3h**:

Figure 3. (h) GORAB KO cells do not phenocopy SCYL1 KO cells. Box plots showing individual EV components and cargoes enrichment in CM of SCYL1 KO/WT and GORAB KO/WT cells, quantified by SILAC-based LC-MS/MS analysis. n=4 biological replicates per condition, indicated by white dots. Data are represented as the Log₂-transformed intensity ratio of KO cell lines over WT; grey line marks the 0 value, indicating no difference to WT. *: p ≤ 0.05; **: p ≤ 0.01; ***: p ≤ 0.001; ****: p ≤ 0.0001 (two-tailed Student's t test between SCYL1 KO and GORAB KO values).

4. Ultrafiltration (UF) was used to isolate EVs in certain cases Fig 4. These cannot be called EVs and rather a very crude prep with contaminants from conditioned media. These data need to be replaced with ultracentrifugation as UF does not provide confidence with the data.

We rewrote this entire section stating that we use ultrafiltration to study the protein content of conditioned medium. Whenever we refer to EVs we used ultracentrifugation as suggested.

5. Fig 4b needs more markers like alix or tsg101 or syntenin and cd63. Was the protein amount normalised to equal cell number or is it total protein? Same for 4h.

We excuse ourselves if not all experimental details were included in the main text or figure legends. We listed these in the materials and methods and moved them to the main text in the revised version. Protein amount was always normalized to cell numbers, which is now stated in the respective figure legends.

As suggested we included several more marker proteins in the old Figure 4h which is now the **new Figure 3h** (see above). We would like to stress that the blots in the old Figure 4b were meant as

validation of our proteomic data presented in Figure 4a. In the respective supplementary table S2 all proteins, i.e. CD63, TSG101 and Alix (PDCD6IP) are listed. If requested we can perform additional blots.

6. Was the particle number normalised against equal live cells in 4e? At the moment it does not look like that? What is the proliferation status of SCYL1 KO cells?

As for the entire figure, also here we used equal amounts of live cells. This was only stated in the methods section and we added respective statements also to the main text.

SCYL1 KO cells appear to grow as WT cells. If requested, we can add respective data.

7. Claim that SCYL1 reduces EV release in Fig 6d lacks QUANTITATIVE data. The Western blot need to be normalised by cell number. In addition NTA data needs to be provided again normalised by equal cell number.

Also this blot was normalized to cell number and we detect proteins of ultracentrifuged CM. We performed these analyses 3 times and added the data as **new Figure 6c-d**. With respect to the NTA analyses, if requested we can redo these analyses.

Figure 6. (c) HA-SCYL1 WT and S754E, but not S754A, rescue SCYL1 KO-induced EV secretion. Western blot analysis of ultracentrifuged CM of SCYL1 KO MCF7 cells expressing HA-SCYL1WT, HA-SCYL1S754A or HA-SCYL1S754E mutants after 24h doxycycline induction. Cells containing a vector only transgene (VC) were used as negative control and treated in the same conditions. To normalize protein contents, equal amounts of cells were used for EV purification. One representative experiment out of n= 3 biological replicates is shown. **(d) Quantification of (c)**, average of n=3 biological replicates. Error bars= SEM. *: p-value ≤ 0.05; (two-tailed Student's t-test between the indicated data points).

8. Additional cell lines need to be used to validate the important observations.

In **new supplementary Figure S1**, we highlight that next to MCF7 cells also A549, HeLa and RPE-1 cells respond with a change in localization of SCYL1 due to mTORC1 inhibition. According to us, this is the most important finding of our manuscript.

9. The use of the TERM exosomes need to be avoided as per MISEV guidelines. The authors can perhaps use small extracellular vesicles (sEV) as they have done 100K. The isolated fraction contains both endosome- and PM-derived EVs so it is wise to name them EVs.

We rewrote the entire paper and only use the term EV as suggested by the reviewer. In addition, the term EV is only used when analyzing ultracentrifuged conditioned medium, otherwise we “just” state that we analyze the protein content of conditioned medium.

10. Proteomics datasets need validation. For instance, several EV proteins are enriched in KO cells – can authors perform Western blot and confirm this.

As mentioned above, **new Figures 3b and supplementary Figure S3c** validate our proteomics data. But we are happy to include more markers if wished.

11. Claims at Page 12 “. These data show that mTORC1-dependent SCYL1 Ser754 phosphorylation is necessary and sufficient to block exosome release, regardless of SCYL1 subcellular localization and the presence or absence of GORAB.” is not supported by the data and is an over claim. No data supporting this has been provided.

We agree that this statement was an overstatement and we removed it.

12. Phosphoantibody generation needs clarity. In Supplementary Fig. 1, were SCYL1 KO cells used? If not, what is the basal endogenous S574 phosphorylation status? The bands detected in S572A/E – are these endogenous levels? The authors say cross reactivity with total SCYL1 while endogenous levels were never mentioned? Because the cells are not clearly mentioned, it is confusing as to whether the authors used MCF7 cells or SCYL1 KO? Also, if KO cells were used – what about the leakage in the DOX system? Controls for these are missing. As authors are not providing more information in text, several questions arise. Can the authors provide more information in the main article throughout – like cells and precise condition etc.

We are sorry for not providing all information. As stated above we rewrote the entire article and added more experimental details to the main text (which were included in the methods section in the original publication).

Old supplemental Figure S1a was performed with KO cells expressing tagged SCYL1 variants. Thus, old Figure S1a highlights the cross-reactivity of the phospho-specific antibody as stated in the main text. Concerning leakiness of the expression system, we have all respective blots of uninduced cells showing that the system is not leaky and could add them if requested.

13. Fig 1 d and e does not correlate well. Western blot shows starvation has more effect on S574-p while 1e shows no difference between starvation and rapamycin (30 min seems almost half between Rapamycin and Starvation in Western blot)

We performed additional experiments and added respective data as **new figure 4** (see above). Minimally 3 blots from biological replicates were performed, one example being shown in **new Figure 4d**. We requested, we can show all three blots in the supplements or upload respective data on a public server.

14. Please mentions cell names throughout – figure 1 description in text and legend has no cell names.

We added respective data as requested.

REVIEWER COMMENTS

Reviewer #1 (Remarks to the Author):

Most of the main concerns have been addressed properly by the authors in the revised version, and the manuscript reads well now. However, I still disagree with the use of breast cancer cell lines to prove the point. Maybe the introduction can be re-written in the light of this? I think that the readers need to have clear why a certain cell model is used and be able to understand the importance of the question in the chosen cell model as well as in the light of other results in the literature. Repeating experiments in other cells lines is appreciated, and I would suggest moving data in supp fig 1 to the main results. Furthermore, other key experiments should be shown in the chosen cell lines as well. Again, the choice of these cell models need to be better justified in the context of the big picture. In conclusion, I think that the question is important but the way to address the question still needs improvements.

Reviewer #2 (Remarks to the Author):

The authors have submitted a fully revised version of the manuscript now addressing all of my concerns. The work suggests a novel hypothesis in the field that should be further tested and expanded in future studies. In my opinion, this does not undercut the importance of the finding but highlights its importance. Therefore, I support the publication of this manuscript.

Reviewer #4 (Remarks to the Author):

I have been asked to do a late stage minimal review of this manuscript because it already underwent revision by other reviewers.

I can't recommend publication because the microscopy data are globally sub-optimal, lacking appropriate analysis and clear quantification methods, and sometimes even not illustrating the claims (how does one measure cell area without physical marker of the plasma membrane?, claims of differential distribution not illustrated by micrographs (sometimes taken with different focus), EM most convincing but just one image...). Concerning the EV part, the methodology is acceptable (UF for differential proteomics, followed by DUC and western blots for validation) if the blots used for Fig. 3 are shown and convincing, if the text is not confusing about the nomenclature and if there is no overinterpretation of the data.

We would like to thank the reviewers and the editor to express their interest in our manuscript. We have addressed all concerns raised by the reviewers and changed the manuscript accordingly. In addition, we addressed all editorial requests and closely followed the “Policies and forms required for resubmission”. Please find below a detailed point-by-point response.

Reviewer #1 (Remarks to the Author):

Most of the main concerns have been addressed properly by the authors in the revised version, and the manuscript reads well now. However, I still disagree with the use of breast cancer cell lines to prove the point. Maybe the introduction can be re-written in the light of this? I think that the readers need to have clear why a certain cell model is used and be able to understand the importance of the question in the chosen cell model as well as in the light of other results in the literature.

We thank the reviewer for raising this concern. We rewrote the introduction highlighting the function of mTORC1 in cancer progression and focusing on breast cancer. In addition, we introduce MCF7 cells as “active secretory cells” and cite the appropriate literature justifying the use of these cells to study the role of mTORC1 and its target SCYL1 in protein secretion:

New text added, page 3, 4 and 7:

“Conversely, hyperactivation of mTORC1 signaling is a frequent event in cancer, with an estimated average occurrence of 70% in all cancer types ⁴. mTORC1 aberrant activation is a tumor driver ⁵, promoting formation, proliferation, and/or invasion, depending on activation timing and cancer type (reviewed in ⁶). Although several pharmacological mTORC1 inhibitors have been developed and tested in anti-cancer studies, resistance phenomena are frequently observed and combinatorial treatments are offered to circumvent these ⁷. Developing efficient therapeutic strategies against mTORC1 overactivation in cancer will therefore require a deeper understanding of the role of this kinase complex and its downstream effectors and functions”

“The N-terminal kinase-like protein SCYL1 maintains Golgi structure and functions by promoting COPI coated vesicle-mediated retrograde transport from Golgi to ER ²⁵⁻²⁸. *Scyl1^{mdf}* mutant mice display motor neuron disorders recapitulating most human amyotrophic lateral sclerosis (ALS) symptoms ²⁹. Moreover, loss of function mutations of human *SCYL1* are associated with multiple recessive hereditary disorders that have been grouped under the term cholestasis, acute liver failure, and neurodegeneration (CALFAN) syndrome ³⁰⁻³⁴. The syndrome has also been associated to recurrent respiratory failure ³⁴. In addition, SCYL1 has been linked to breast cancer progression, but its precise oncogenic function is disputed ^{35, 36}. Given its subcellular localization and the affected tissues, these observations suggest an important role for SCYL1 in the function of active secretory cells.”

“MCF-7 cells are considered poorly aggressive with a low metastatic potential. They conserved their secretory capacity, since their culture in egg white stimulates the development of acini and duct-like structures that support the secretion of beta-casein (reviewed in ³⁸).”

Repeating experiments in other cells lines is appreciated, and I would suggest moving data in suppl fig 1 to the main results. Furthermore, other key experiments should be shown in the chosen cell lines as well. Again, the choice of these cell models need to be better justified in the context of the big picture. In conclusion, I think that the question is important but the way to address the question still needs improvements.

As suggested we moved data from suppl. Figure 1 to the main Figure 1 and highlight that upon mTORC1 inhibition SCYL1 localization changes in HeLa and A549 cells as in MCF-7 cells. In the new suppl. Figure 1 we still show that this phenomenon also takes place in RPE-1 cells. Thus, we tested four cell lines in total, all behaving similarly (three showing significant changes and one showing a similar trend). We added new, fine-grained quantifications using cell and nuclear volume reconstruction (Imaris Microscopy Image Analysis Software, cell body detection tool) to strengthen our observations.

New Figures 1a and 1b:

Figure 1: mTORC1 inhibition leads to SCYL1 redistribution to cytoplasmic locations including early/late endosome compartments.

(a) SCYL1 localizes to cell periphery upon mTORC1 inhibition in various epithelial cell lines. Endogenous SCYL1 puncta localize further away from nuclei upon rapamycin treatment. Representative fluorescence microscopy images from MCF-7, HeLa and A549 cells in untreated (DMEM) and Rapamycin-treated (Rapa, 3 h, 100 nM) conditions. Cells were immunostained with antibodies against SCYL1 (green). DNA was stained with Hoechst reagent. One representative image out of $n > 3$ biological replicates is shown. Scale bar = 10 μm . Wide field images are available in supplementary Figure 1. **(b)** Quantification of SCYL1 puncta distribution shown in (a). Distances of individual SCYL1 puncta from the closest nuclear envelope were averaged per image and their relative distribution is shown. An unpaired two-tailed Student's t-test was used to compare DMEM and Rapa-treated SCYL1 puncta populations in each cell line: ****: $p \leq 0.0001$ (all p-values are available in the Source data materials). $n = 3$ replicates, with a minimum of 10 cells measured per replicate. Error bars: SEM, ns: not significant.

Reviewer #2 (Remarks to the Author):

The authors have submitted a fully revised version of the manuscript now addressing all of my concerns. The work suggests a novel hypothesis in the field that should be further tested and expanded in future studies. In my opinion, this does not undercut the importance of the finding but highlights its importance. Therefore, I support the publication of this manuscript.

We thank the reviewer for her/his interest and positive feedback.

Reviewer #4 (Remarks to the Author):

I have been asked to do a late stage minimal review of this manuscript because it already underwent revision by other reviewers.

I can't recommend publication because the microscopy data are globally sub-optimal, lacking appropriate analysis and clear quantification methods, and sometimes even not illustrating the claims (how does one measure cell area without physical marker of the plasma membrane?, claims of differential distribution not illustrated by micrographs (sometimes taken with different focus), EM most convincing but just one image...).

We thank the reviewer for her/his critical feedback. We changed all microscopy data as well as respective data quantifications. Briefly, all micrographs show now a comparable number of cells using the same magnification. Wide-field data has been moved to supplements.

As suggested, we specifically stained the plasma membrane using TJP3 or CD324/E-cadherin. In all cases, however, the 3D reconstruction of cells was inaccurate, likely due to the non-homogenous distribution of the two proteins in the plasma membrane (**Figure 1 for review only**). We assume that the nature of MCF-7 cells does not support proper membrane staining with these commonly used markers.

Figure 1 for review only:

Figure 1 for review only: TJP3 and E-CADHERIN cell membrane markers do not support proper MCF7 cell boundary definition.

WT MCF7 cells were immunostained with the indicated antibodies targeting plasma membranes using cell junction proteins and imaged as described in material and methods (left panels). Right panel: zoom-in pictures showing multiple missing boundaries (white arrows) not detected by the markers, or tight junctions crossing intracellular structures (yellow arrows). Scale bars: 10 μm .

To address these problems and to avoid examiner-caused bias we turned to a software-based solution. We determined cell volumes using 3D cell reconstructions by the Imaris Microscopy Image Analysis Software (Bitplane), which automatically identifies 3D cell boundaries using signals of cytosolic marker proteins. For closely touching cells, Imaris Cell algorithm calculates all signal distances from nuclei and computes the best solution possible, i.e. the closest distance to a specific nucleus, taking one nucleus per cell as basic characteristic. For quantifications we used 3D-reconstructed cell, nuclear, and organelle/vesicle volumes. All analyses were manually inspected to ensure proper cell segmentation. Incomplete cells, cells on image borders, as well as technically damaged cells, were excluded from further analyses.

New figures 1a and 1b (see above, comments to reviewer 1).

New figures 2 b-c, f-h:

Figure 2: SCYL1 regulates Golgi structure and endolysosomal trafficking.

(b) SCYL1 KO cells display an enlarged Golgi. Representative fluorescence microscopy images from WT and SCYL1 KO MCF-7 cells (clone KO1), immunostained for endogenous SCYL1 (green) and Golgi GOLGIN-97/GOLGA1 (red). Scale bar, 10 μm . **(c)** Quantification of (b). The ratio between the Golgi apparatus volume, labeled with GOLGIN97/GOLGA1 antibodies, and the total cellular volume, detected with the cell body detection tool (Imaris), was computed in individual cells. All results for WT and SCYL1 KO cells were assembled in a violin plot (median: central white dot;

interquartile range: thick black line; full range: thin black line). Single cell values are indicated by white small dots. An unpaired two-tailed Student's t-test was used to compare WT and SCYL1 KO values: *: $p \leq 0.05$; **: $p \leq 0.01$; ***: $p \leq 0.001$; ****: $p \leq 0.0001$ (all p-values are available in the Source data materials). $n=3$ biological replicates with at least 36 individual cells measured per replicate. Error bar= SEM. **(f)** Loss of SCYL1 leads to a perturbation of early and late endosome localization. Representative fluorescence microscopy images from WT and SCYL1 KO MCF-7 cells in normal growth conditions (DMEM). Cells were immunostained for early endosome (RAB5) and late endosome (RAB7) markers. The merged picture also shows DNA stained by Hoechst reagent. Representative pictures of $n=3$ replicates are shown. Scale bar = 10 μm . **(g)** Loss of SCYL1 leads to a perturbation of lysosome subcellular distribution. Representative fluorescence microscopy images from WT and SCYL1 KO MCF-7 cells in normal growth conditions (DMEM). Cells were immunostained for lysosomal marker LAMP2. The merged picture also shows DNA stained by Hoechst reagent. Shown are representative pictures of minimally two technical per two biological replicates ($n=4$). Scale bar, 10 μm . **(h)** Quantification of (f-g). The distance from nuclear envelope to RAB5, RAB7 and LAMP2 foci was computed and averaged per individual cell. All results for WT and SCYL1 KO cells were assembled in a violin plot (median: central white dot; interquartile range: thick black line; full range: thin black line). Single cell values are indicated with white small dots. An unpaired two-tailed Student's t-test was used to compare WT and SCYL1 KO populations; *: $p \leq 0.05$; **: $p \leq 0.01$; ***: $p \leq 0.001$; ****: $p \leq 0.0001$ (all p-values are available in the Source data materials). 10 individual cells were individually measured within three technical replicates per two biological replicates ($n=6$). Error bar, SEM

New figures 5 b-c:

Figure 5: Ser754 phosphorylation modulates SCYL1 functions in Golgi structure maintenance and EV secretion.

(b) SCYL1 Ser754 phosphorylation maintains Golgi architecture. Contrary to HA-SCYL1^{WT} and HA-SCYL1^{S754E} proteins, which localize closer to the Golgi and around the nucleus, the HA-SCYL1^{S754A} phospho-null localizes to the cell periphery. In addition, the enlarged Golgi observed in SCYL1 KO

cells is rescued by HA-SCYL1^{WT} and HA-SCYL1^{S754E}, but not by HA-SCYL1^{S754A}. Representative fluorescent microscopy pictures of SCYL1 KO MCF-7 cells re-expressing HA-SCYL1^{WT}, HA-SCYL1^{S754A} or HA-SCYL1^{S754E} upon 24 h doxycycline induction. Cells were immunostained for SCYL1 (green) and Golgi apparatus marker GOLGIN-97/GOLGA1 (red). Blue channel: Hoechst reagent, marking DNA. n=3 biological replicates. **(c)** Quantification of **(b)**. The ratio between the Golgi apparatus volume, labeled with GOLGIN97/GOLGA1 antibodies, and the total cellular volume, detected with the cell body detection tool (Imaris), was computed in individual cells. All results for SCYL1 KO MCF-7 cells re-expressing HA-SCYL1^{WT}, HA-SCYL1^{S754A} or HA-SCYL1^{S754E} for 24 h or 48 h were assembled in a violin plot (median: central white dot; interquartile range: thick black line; full range: thin black line). Single cell values are indicated with small white dots. An unpaired two-tailed Student's t-test was used (all p values are detailed in Source data materials); *, p ≤ 0.05; **, p ≤ 0.01; ***, p ≤ 0.001; ****, p ≤ 0.0001). ≥30 cells were measured amongst technical triplicates of two biological replicates (n=6). Error bars, SEM.

New figures 6 a-c:

Figure 6: mTORC1-dependent SCYL1 Ser754 phosphorylation regulates subcellular endosome distribution and EV secretion. (a-b) Expression of SCYL1^{S754E} complements SCYL1 KO effect on

endosome distribution. Representative fluorescent microscopy pictures of SCYL1 KO MCF-7 cells expressing HA-SCYL1^{WT}, HA-SCYL1^{S754A} or HA-SCYL1^{S754E} variant upon 24 h (for RAB5 and RAB7) and 36 h doxycycline induction (for LAMP2), respectively. Cells were immunostained for early endosome marker RAB5 and late endosome marker RAB7 (a), or for late endosome marker RAB7 and lysosomal marker LAMP2 (b). The merged pictures also show DNA stained by Hoechst reagent. Representative pictures of n=3 replicates are shown. Scale bar = 10 μ m. **(c)** Quantification of (a-b). The distance from nuclear envelope to RAB5, RAB7 and LAMP2 foci was computed and averaged per individual cell. All results for SCYL1 KO MCF-7 cells expressing HA-SCYL1^{WT}, HA-SCYL1^{S754A} or HA-SCYL1^{S754E} variants upon 24 h (for RAB5 and RAB7) and 36 h doxycycline induction (for LAMP2), were assembled in a violin plot (median: central white dot; interquartile range: thick black line; full range: thin black line). Single cell values are indicated with white small dots. An unpaired two-tailed Student's t-test was used for comparison; *: $p \leq 0.05$; **: $p \leq 0.01$; ***: $p \leq 0.001$; ****: $p \leq 0.001$ (all p-values are available in the Source data materials). ≥ 20 cells were measured amongst technical triplicates of two biological replicates (n=6). Error bars, SEM.

Regarding EM data, we included a second set of electron micrographs in figure 5 and additional two sets in **new supplementary figure 5a**.

Figure 5: SCYL1 Ser754 phosphorylation mediates its functions in Golgi structure maintenance and EV secretion. (a) Golgi structure depends on SCYL1 Ser754 phosphorylation. Two representative TEM pictures of MCF-7 SCYL1 KO cells re-expressing either HA-SCYL1^{WT}, HA-

SCYL1^{S754A} or HA-SCYL1^{S754E} upon 24 h doxycycline induction are shown. D, degradative compartment; E, endosome; G, Golgi apparatus; M, mitochondria; N, nucleus; PM, plasma membrane. N: Nucleus; G: Golgi apparatus; PM: Plasma membrane; #: lipid droplets. Scale bar: 500 nm. Additional pictures are available in supplementary Figure 5.

Concerning the EV part, the methodology is acceptable (UF for differential proteomics, followed by DUC and western blots for validation) if the blots used for Fig. 3 are shown and convincing, if the text is not confusing about the nomenclature and if there is no overinterpretation of the data.

We thank the reviewer for the positive feedback. We checked the MISEV2018 guidelines and followed the nomenclature recommendations described in the document. The extracellular vesicles detected and characterized by us fulfil all characteristics of small EVs. We added supplementary wide-field TEM pictures and TEM pictures with EV diameter measurements to complete our characterization (**new supplementary Figure 3a-b**). We also added a **new supplementary table 2B** in which we highlight all detected marker proteins and their classes according to the MISEV2018 guidelines. In addition, we use a congruent nomenclature throughout the entire manuscript and avoid any overinterpretation of our results.

New supplementary Figure 3a-b

Supplementary Figure 3: SCYL1 KO cells secrete small EVs in a GORAB-independent manner. (a-b) SCYL1 KO cells secrete small EVs. (a) wide-field Scanning Electron Microscopy (SEM) pictures and corresponding zoom regions as displayed in Figure 3D. Scale bar: 1 μ m. **(b)** measurement of

extracellular vesicles in WT (upper panel) and SCYL1 KO (lower panel) conditioned extracellular medium. Ultracentrifuged, concentrated EVs from equal amounts of each cell line were laid on silicon wafers, allowing for clearer pictures and precise measurement of their diameter. WT EVs are in the medium/large EV diameter range (>200 nm), whereas SCYL1 KO EVs show a diameter <200 nm, corresponding to the small EV category. Scale bar: 1 μ m.

REVIEWERS' COMMENTS

Reviewer #1 (Remarks to the Author):

All my concerns have been addressed by the authors.